 **eLIFE**

# Neural pattern change during encoding of a narrative predicts retrospective duration estimates

**Olga Lositsky[1]\*, Janice Chen[1], Daniel Toker[2], Christopher J Honey[3], Michael Shvartsman[1], Jordan L Poppenk[4], Uri Hasson[1,5], Kenneth A Norman[1,5]\***

[1]Princeton Neuroscience Institute, Princeton University, Princeton, United States; [2]Helen Wills Neuroscience Institute, University of California, Berkeley, Berkeley, United States; [3]Department of Psychology, University of Toronto, Toronto, Canada; [4]Department of Psychology, Queen's University, Kingston, Canada; [5]Department of Psychology, Princeton University, Princeton, United States

**Abstract** What mechanisms support our ability to estimate durations on the order of minutes? Behavioral studies in humans have shown that changes in contextual features lead to overestimation of past durations. Based on evidence that the medial temporal lobes and prefrontal cortex represent contextual features, we related the degree of fMRI pattern change in these regions with people's subsequent duration estimates. After listening to a radio story in the scanner, participants were asked how much time had elapsed between pairs of clips from the story. Our ROI analyses found that duration estimates were correlated with the neural pattern distance between two clips at encoding in the right entorhinal cortex. Moreover, whole-brain searchlight analyses revealed a cluster spanning the right anterior temporal lobe. Our findings provide convergent support for the hypothesis that retrospective time judgments are driven by 'drift' in contextual representations supported by these regions.

**\*For correspondence:** lositsky@ princeton.edu (OL); knorman@ princeton.edu (KAN)

**Competing interests:** The authors declare that no competing interests exist.

## Introduction

Imagine that you are at the bus stop when you run into a colleague and the two of you become engrossed in a conversation about memory research. After a few minutes, you realize that the bus still has not arrived. Without looking at your watch, you have some sense of how long you have been waiting. Where does this intuition come from?

Estimation of durations lasting a few seconds has been probed in the neuroimaging, neuropsychology and neuropharmacology literatures (see *Wittmann, 2013*, for a review). On the other hand, the neural mechanisms underlying time perception on the scale of minutes have remained unexplored. This is particularly true of *retrospective* judgments, where individuals experience an interval without paying attention to time and must subsequently estimate the interval's duration. In such cases, individuals must rely on information stored in memory to estimate duration. How is this accomplished?

Memory scholars have long posited that the same contextual cues that help us to retrieve an item from memory can also help us determine its recency. According to extant theories of context and memory (see *Manning et al., 2014*, for a review), *mental context* refers to aspects of our mental state that tend to persist over a relatively long time scale; this encompasses our representation of slowly-changing aspects of the external world (e.g., what room we are in) as well as other slowly-changing aspects of our internal mental state (e.g., our current plans). Crucially, these theories posit that slowly-changing contextual features can be episodically associated with more quickly-changing

**eLife digest** How do humans judge how much time has passed during daily life, such as when waiting for the bus? Psychology studies have shown that people remember events to have lasted longer when more changes occurred during that time period. These changes can occur either in the environment (such as changes in location) or in the individual's internal state (such as changes in goals and emotions).

Brain activity changes from moment to moment. Lositsky et al. hypothesized that when patterns of activity in a person's brain change a lot across an interval of time, that person will judge that a long time has passed. On the other hand, if brain activity changes less over that interval, individuals will judge that less time has passed.

Some regions of the brain are sensitive to information that unfolds over several minutes; many of these regions are vital for forming memories of episodes from our lives. Using a technique called functional magnetic resonance imaging (fMRI), Lositsky et al. specifically looked at the activity of these regions while volunteers listened to a 25-minute radio drama. Afterwards, the volunteers listened to clips from different events in the story and judged how much time passed between those events.

Even though each pair of audio clips occurred exactly two minutes apart in the original story, people's time judgments were strongly influenced by how many scene changes happened in the story between the two clips. In a part of the brain called the right anterior temporal lobe – and especially in a region of it called the entorhinal cortex – Lositsky et al. found that brain activity changed more when audio clips were judged to be further apart in time. Activity in this region fluctuated more slowly overall than in the rest of the brain. This could mean that it combines sensory information (about images, sounds, smells and so on) across minutes of time, in order to form a representation of the current situation.

Future research could focus on several unanswered questions. Exactly which environmental and internal changes influence our perception of time? What form does this information take in the entorhinal cortex? Studies show that the entorhinal cortex contains "grid cells" that track our location in space. Could these cells also help judge the passage of time?

aspects of the world (e.g., stimuli that appear at a particular moment in time; *Mensink and Raaijmakers, 1988*; *Howard and Kahana, 2002*).

*Bower (1972)* first proposed that we could determine how long ago an item occurred by comparing our current context with the context associated with the remembered item. The similarity of these two context representations would reflect their temporal distance, with more similar representations associated with events that happened closer together in time. Thus, a slowly varying mental context could serve as a temporal tag (*Polyn and Kahana, 2008*). In parallel, researchers in the domain of retrospective time estimation have shown that the degree of context change is a better predictor of duration judgments than alternative explanations, such as the number of items remembered from the interval (*Block and Reed, 1978*; *Block, 1990*, *1992*). Indeed, changes in task processing (*Block and Reed, 1978*; *Sahakyan and Smith, 2014*), environmental context (*Block, 1982*), and emotions (*Pollatos et al., 2014*), as well as event boundaries (*Poynter, 1983*; *Zakay et al., 1994*; *Faber and Gennari, 2015*), lead to overestimation of past durations.

In our study, we set out to obtain neural evidence in support of the hypothesis that mental context change drives duration estimates. Specifically, we hypothesized that, in brain regions representing mental context, the degree of neural pattern change between two events (operationalized as change in multi-voxel patterns of fMRI activity) should predict participants' estimates of how much time passed between those events.

Extensive prior work has implicated the medial temporal lobe (MTL) and lateral prefrontal cortex (PFC) in representing contextual information (*Polyn and Kahana, 2008*; for reviews of MTL contributions to representing context, see *Eichenbaum et al., 2007*, and *Ritchey and Ranganath, 2012*; for related computational modeling work, see *Howard and Eichenbaum, 2013*). In keeping with our hypothesis, multiple studies have obtained evidence linking neural pattern change in these regions

to temporal memory judgments. *Manns et al. (2007)* recorded from rat hippocampus during an odor memory task; they found that greater change in hippocampal activity patterns between two stimuli predicted better memory for the order in which the stimuli occurred. In the human neuroimaging literature, *Jenkins and Ranganath (2010)* found that the degree to which activity patterns in rostrolateral prefrontal cortex changed during the encoding of a stimulus predicted better memory for the temporal position of that stimulus in the experiment. *Jenkins and Ranganath (2016)* also showed that greater pattern distance between two stimuli at encoding in the hippocampus, medial and anterior prefrontal cortex predicted better order memory. Only one study has directly related neural pattern drift to judgments of elapsed time in humans: *Ezzyat and Davachi (2014)* found that patterns of fMRI activity in left hippocampus were more similar for pairs of stimuli that were later estimated to have occurred closer together in time, despite equivalent time passage between all pairs (a little less than a minute).

While the *Ezzyat and Davachi (2014)* study provides support for our hypothesis, it has some limitations. First, in *Ezzyat and Davachi (2014)*, participants estimated the temporal distance of stimuli that were linked to their contexts in an artificial way (by placing pictures of objects or famous faces on unrelated scene backgrounds); it is unclear whether these results will generalize to more naturalistic situations where events are linked through a narrative. Second, since participants performed the temporal memory test after each encoding run, they were not entirely naïve to the manipulation. Knowing that they would have to estimate durations between stimuli could have changed participants' strategy and enhanced their attention to time (for evidence that estimating time prospectively engages different mechanisms, see *Hicks et al., 1976*, and *Zakay and Block, 2004*). In the current study, we sought to address the above issues by eliciting temporal distance judgments for pairs of events that had occurred several minutes apart and that were embedded in the context of a rich naturalistic story; participants listened to the entire story before being informed about the temporal judgment task.

Based on the studies reviewed above, we predicted that neural pattern drift in medial temporal and lateral prefrontal regions might support duration estimation. In our study, we examined these regions of interest (ROIs), as well as a broader set of regions that have been implicated in fMRI studies of time estimation, including the inferior parietal cortex, putamen, insula and frontal operculum (see *Box 1* for a review). In addition to the ROI analysis, which examined activity patterns in masks that were anatomically defined, we performed a searchlight analysis, which examined activity patterns within small cubes over the whole brain.

## Box 1. fMRI literature on prospective time estimation.

As noted in the main text, only one study (*Ezzyat and Davachi, 2014*) has used fMRI to study retrospective estimation of time intervals lasting more than a few seconds. The vast majority of fMRI studies of time estimation have used prospective tasks, in which participants are asked to deliberately track the duration of a short stimulus or compare the duration of two stimuli. Such studies have repeatedly shown that activity in the putamen, insula, inferior frontal cortex (frontal operculum), and inferior parietal cortex increases as participants pay more attention to the duration of stimuli, as opposed to another time-varying attribute (*Coull, 2004*; *Coull et al., 2004*; *Livesey et al., 2007*; *Wiener et al., 2010*; *Wittmann et al., 2010*). *Dirnberger et al. (2012)* showed that greater activity in the putamen and insula during encoding of aversive emotional pictures predicted better subsequent memory for those pictures, but only when their duration was overestimated relative to neutral images. This suggests that the putamen and insula might mediate the relationship between enhanced processing for emotional stimuli and subjective time dilation. Given the established role of these regions in time processing (albeit of a different sort), we included these regions in the set of *a priori* ROIs for our main fMRI analysis.

Participants were scanned while they listened to a 25-minute science fiction radio story. Outside the scanner, they were surprised with a time perception test, in which they had to estimate how much time had passed between pairs of auditory clips from the story. Controlling for objective time, we found that the degree of neural pattern distance between two clips at the time of encoding predicted how much time an individual would later estimate passed between them. The effect was significant in the right entorhinal cortex ROI. Extending the anatomical analysis to all masks in cortex revealed an additional effect in the left caudal anterior cingulate cortex (ACC). Moreover, whole-brain searchlight analyses yielded significant clusters spanning the right anterior temporal lobe. Our results suggest that patterns of neural activity in these regions may carry contextual information that helps us make retrospective time judgments on the order of minutes.

## Results

### Behavioral results

Participants were sensitive to the duration of story intervals

*Figure 1* shows the experimental design, which consisted of an fMRI session, followed immediately by a behavioral session. After listening to a 25-min radio story in the scanner, participants were asked how much time had passed between 43 pairs of clips from the story. In actuality, 24 of the clip pairs had been presented 2 minutes apart in the story, while 19 of the clip pairs had been presented 6 minutes apart in the story (participants were not informed of this). Participants were able to estimate the duration of experienced minutes-long intervals far above chance, albeit with substantial intra- and inter-individual variability. On average, across participants, the 6-min intervals ($M$=5.70 min, $SD$=3.06) were judged to be significantly longer than the 2-min intervals ($M$=3.69 min, $SD$=1.96), $t(17) = 5.20$, $p<10^{-4}$ (see *Figure 2A*).

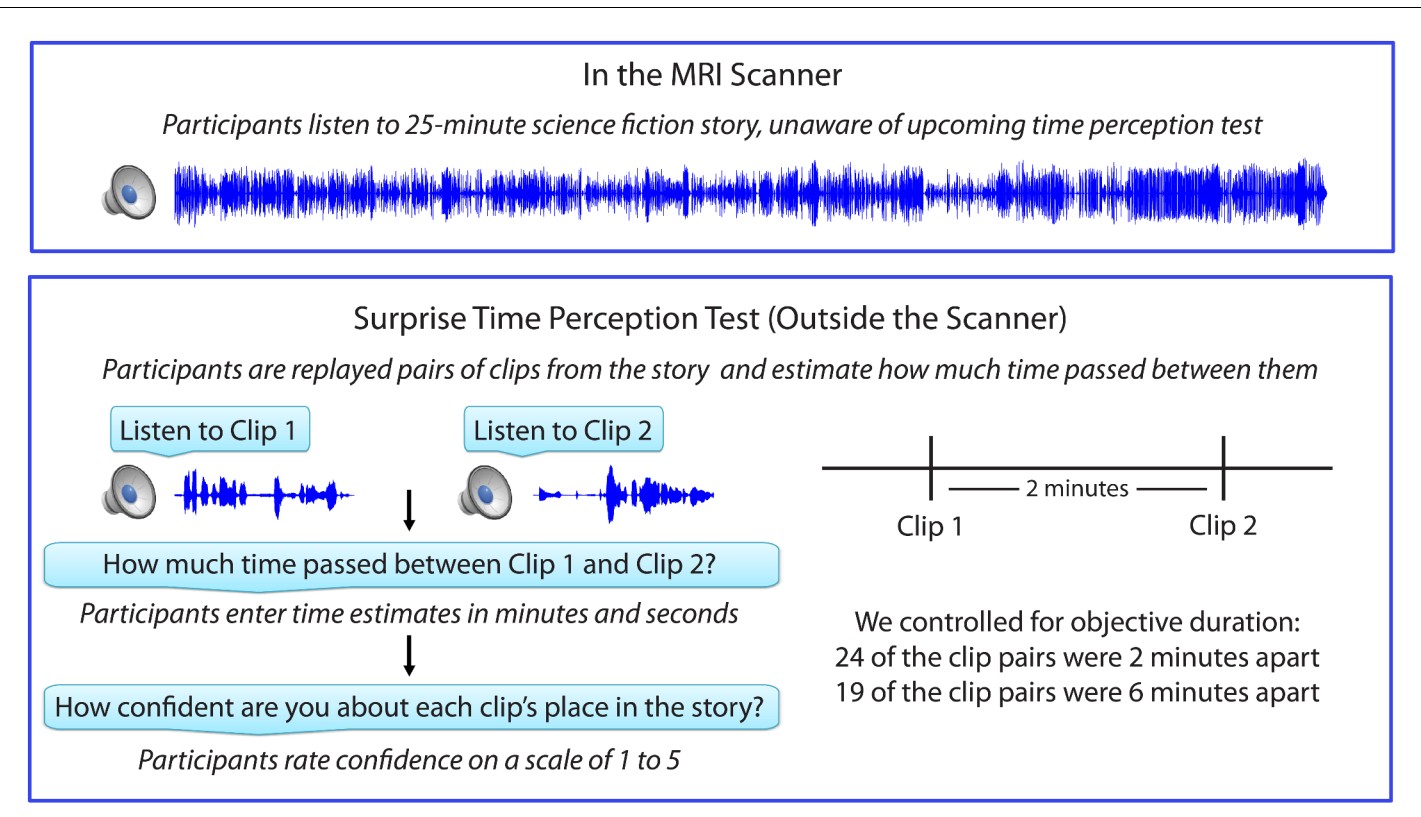

**Figure 1.** Experimental design.

As described in the Materials and methods (see *Removing low-confidence intervals*), participants also provided confidence ratings reflecting their certainty about each clip's place in the story. Based on this measure, we grouped each participant's duration estimates into high-confidence and low-confidence intervals. To verify that participants were better at distinguishing 6-min intervals from 2-min intervals when they were confident, we calculated the difference between the mean duration estimates for 6-min intervals and the mean duration estimates for 2-min intervals for every participant. The difference score was significantly higher for high-confidence intervals (*M*=2.43, *SD*=1.82) than for all intervals (*M*=2.01, *SD*=1.64), *t*(17)=2.33, *p*=0.0324. Thus, participants were significantly more accurate at estimating an interval's duration when they confidently remembered the temporal position of both clips delimiting that interval in the story (see *Figure 2B*).

For a given interval duration, some intervals were consistently judged to be longer than other intervals across participants, although the actual amount of elapsed time was held constant. To test the reliability of duration estimates across participants, we split the subjects randomly into two groups, averaged the duration estimates within each group, and correlated the two averages with each other. We repeated this procedure 1000 times to ensure that we sampled a variety of group splits. The average correlation between the two groups was 0.64 (*SD*=0.09) for 2-min intervals and 0.54 (*SD*=0.15) for 6-min intervals (see *Figure 2—figure supplement 1*). This analysis suggests that features of the story made some intervals appear consistently shorter and other intervals appear consistently longer across participants.

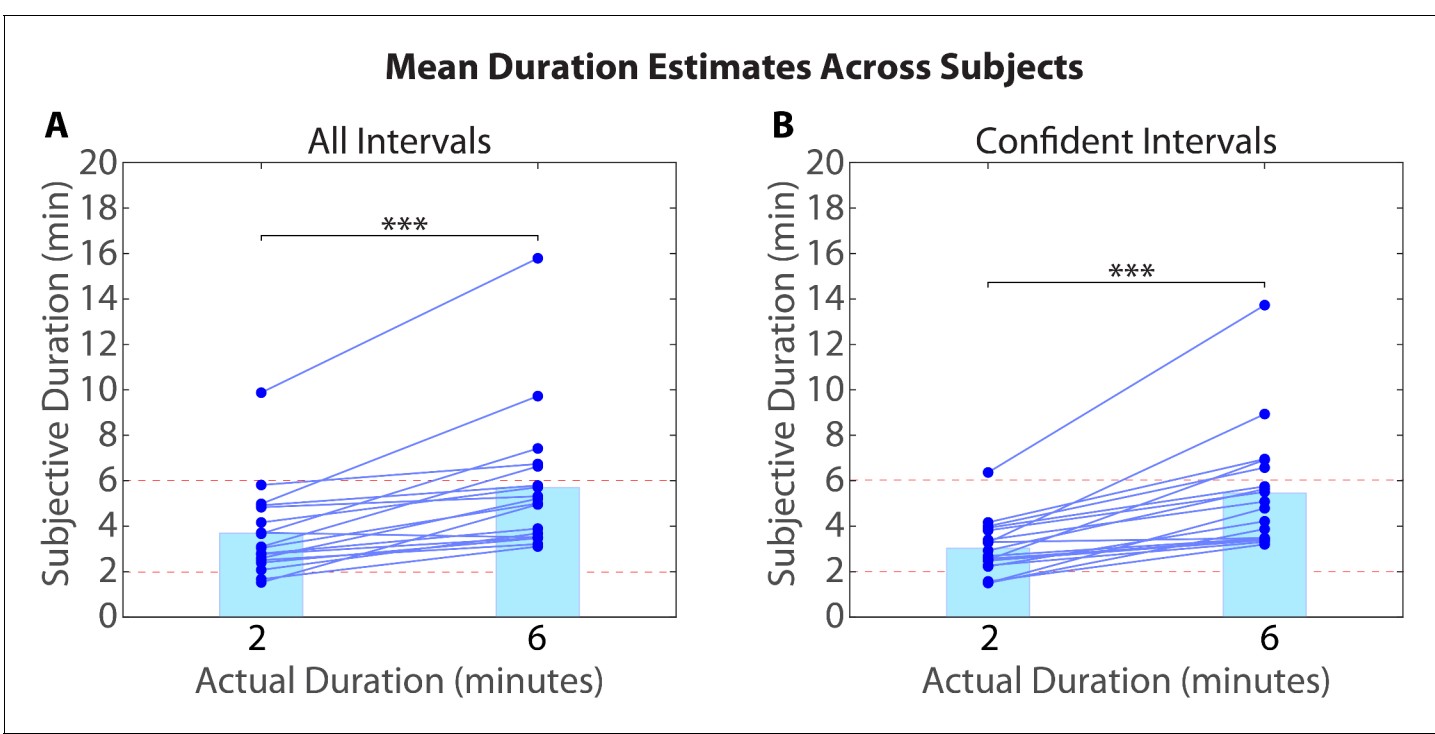

**Figure 2.** Mean duration estimates for all intervals (A) and confident intervals (B) as a function of their actual duration. Each blue circle represents the mean duration estimate for an individual participant within a given interval duration (2 or 6 min). The blue bar heights represent the global means for 2 and 6-min intervals across intervals and participants.

The following source data and figure supplement are available for figure 2:

**Source data 1.** Duration estimates and confidence ratings for all participants and intervals.

**Figure supplement 1.** Reliability of duration estimates across participants.

## Duration estimates are influenced by memory of the story

We found that participants estimated six-minute intervals to be significantly longer than two-minute intervals (*Figure 2*), and that some intervals in the story tended to be systematically over-estimated by participants (*Figure 2—figure supplement 1*). However, it is possible that participants could judge the temporal distance between two clips purely based on the similarity between them (e.g. Are the same characters speaking? Is the background music the same? Is the topic of conversation similar?).

To ensure that participants were using their memory of the story to judge temporal distance, we ran a control experiment in which 17 participants who had never heard the story were given the exact same memory test. They were asked to try to estimate the amount of time that had elapsed between each pair of clips during the original telling of the story. During debriefing, participants reported making duration estimates based on the perceptual and semantic similarity between the two clips (e.g., which character voices were present, which background music was playing, the topic of conversation).

We found that naïve participants estimated 6-min intervals (*M*=6.21 min, *SD*=1.91) to be longer than 2-min intervals (*M*=5.63 min, *SD*=1.74; *t*(16)=2.62, *p*=0.019), suggesting that the similarity between two clips carried some information about the temporal distance between them. However, naïve participants were significantly less accurate at distinguishing 6-min intervals from 2-min intervals than our original participants who had heard the story. To quantify this, we calculated the difference between the mean duration estimates for 6-min intervals and the mean duration estimates for 2-min intervals for every participant (exactly as above). The difference score was significantly higher for our original participants (*M*=2.01 min, *SD*=1.64 min) than for naïve participants (*M*=0.59 min, *SD*=0.91 min), *t*(26.86)=−3.22, *p*<0.005. Thus, having memory of the story enabled our participants to estimate durations with significantly higher accuracy.

We hypothesized that both our original participants and the naïve participants would use consistent strategies to estimate the temporal distance between two clips, but that these strategies would differ across groups. If this is the case, duration estimates should be more correlated across participants within groups than across participants between groups. The correlation in duration estimates across participants within a group (see Materials and methods) was as strong for naïve participants (*M*=0.43, *SD*=0.18, 95% CI [0.40, 0.56]) as for our original participants (*M*=0.43, *SD*=0.25, 95% CI= [0.37, 0.58]), suggesting that both groups used a consistent strategy to estimate the distance between two clips. When we correlated duration estimates from our original group of participants with those of our naïve participants, we found that the between-group correlations (*M*=0.18, *SD*=0.22, 95% CI=[0.04, 0.28]) were significantly above 0, suggesting that a component of the original duration estimates was influenced by the similarity in content between clips. However, the between-group correlations were significantly lower than the within-group correlations (*p*<0.0001, as assessed by a permutation test described in the *Materials and methods*). In other words, there is a reliable component of our original participants' behavior that cannot be captured by accounting for the perceptual and semantic similarity between clips. In summary, having memory of the story induced a qualitatively different pattern of behavior and produced significantly more accurate duration estimates.

## Correlation between number of event boundaries and duration estimates

To gain additional evidence that duration estimates were related to contextual change, we looked at the correlation between estimated duration and the number of event boundaries in the interval between the clips. The number of intervening event boundaries can be viewed as a proxy for contextual change, insofar as event boundaries often encompass changes in scene, characters and conversation topic (*Kurby and Zacks, 2008*; *Zacks et al., 2009*). As reviewed in the Introduction, numerous studies have found a relationship between changes in contextual features during an interval and duration estimates for that interval.

A separate group of participants (n=9) listened to the story and was asked to press a button every time they felt an event boundary was occurring. These data were then averaged across participants to obtain the mean number of event boundaries inside each two-minute interval. We found that the mean number of boundaries in an interval was significantly correlated with the mean duration estimates from our original experiment (*r* = 0.49, 95% CI [0.27, 0.57]; *Figure 3*). This suggests

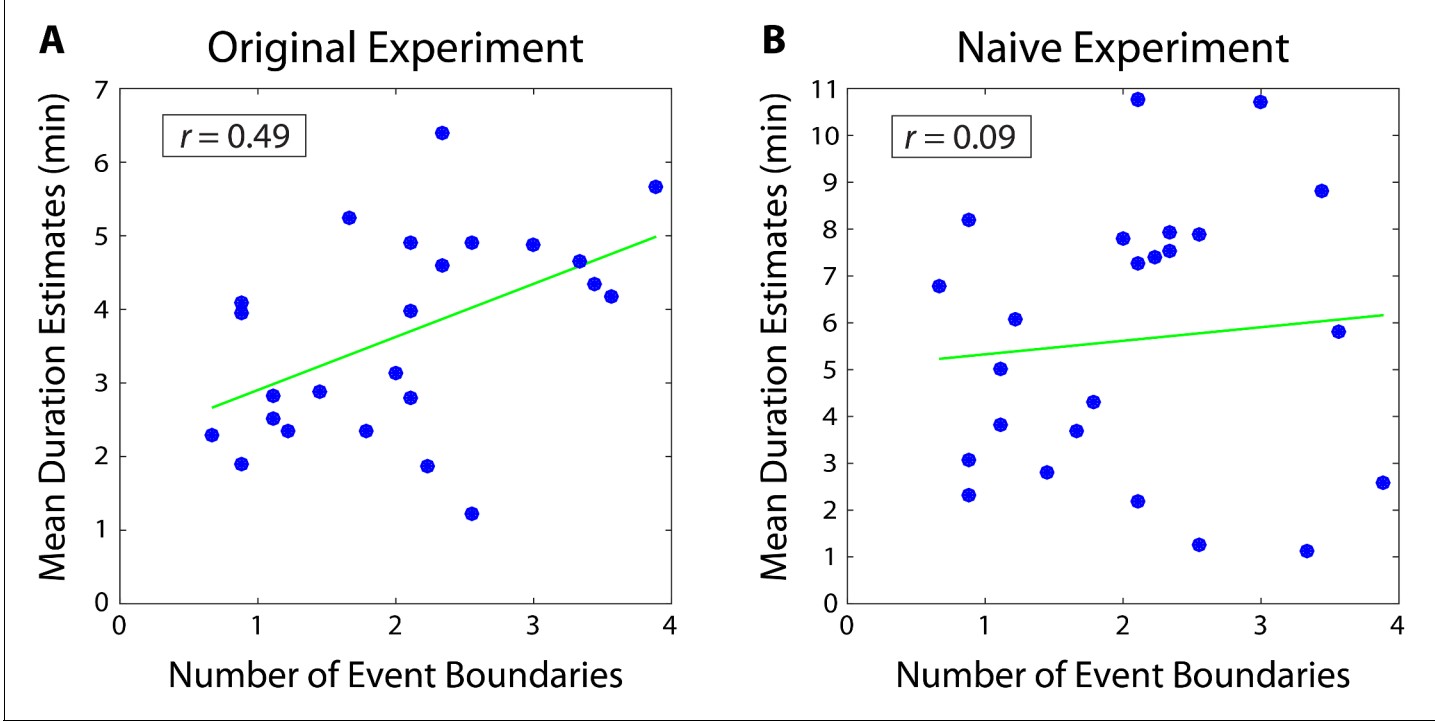

**Figure 3.** Mean duration estimates for 2-minute intervals as a function of the number of event boundaries in each interval. The number of event boundaries in an interval predicted retrospective duration estimates in our original experiment (**A**), but did not significantly predict duration estimates of naïve participants (**B**) who had never heard the story. This suggests that the number of contextual changes between two clips influenced temporal distance judgments significantly more when the content of the story between the two clips could be recalled.

The following source data is available for figure 3:

**Source data 1.** Mean number of event boundaries and mean duration estimates from both original and naïve participants.
**Source data 2.** Duration estimates from the naïve experiment, including both 2 and 6-min intervals.

that our participants' retrospective duration estimates were influenced by the number of contextual changes that had occurred during an interval.

However, it is important to note that the number of event boundaries between two clips also influences the perceptual and semantic similarity between them (e.g., clips from the same scene might sound more similar than clips from different scenes). Thus, our participants' duration estimates could correlate with the number of event boundaries, even if they were basing their estimates purely on the perceptual similarity between clips. To explore this possibility, we tested whether the number of event boundaries would correlate with duration estimates from naïve participants, who could *only* judge temporal distance based on the similarity between clips, given that they had never heard the story.

Importantly, we found that the number of event boundaries in an interval did not significantly correlate with duration estimates of naïve participants ($r=0.09$, 95% CI [−0.05, 0.21]; *Figure 3*). Of course, we cannot definitively prove the null hypothesis that naïve duration estimates do not correlate with the number of event boundaries. However, the correlation between the number of boundaries and duration estimates was significantly higher for our original participants than for naïve participants ($r_{diff}=0.40$, 95% CI [0.15 0.56]). In other words, duration estimates from participants who remembered the story were significantly more correlated with the number of contextual changes between two clips than duration estimates from participants who were judging temporal distance based merely on the similarity between the two clips. This suggests that the number of event boundaries carries information about temporal context that is not contained within the clips alone, and

that our original participants' estimates were influenced by their memory of this contextual information.

## fMRI results

We tested whether BOLD pattern change between two clips correlated with temporal distance estimates, using both ROI and whole-brain searchlight analyses. Each type of analysis was performed both within-participants across intervals and within-intervals across participants.

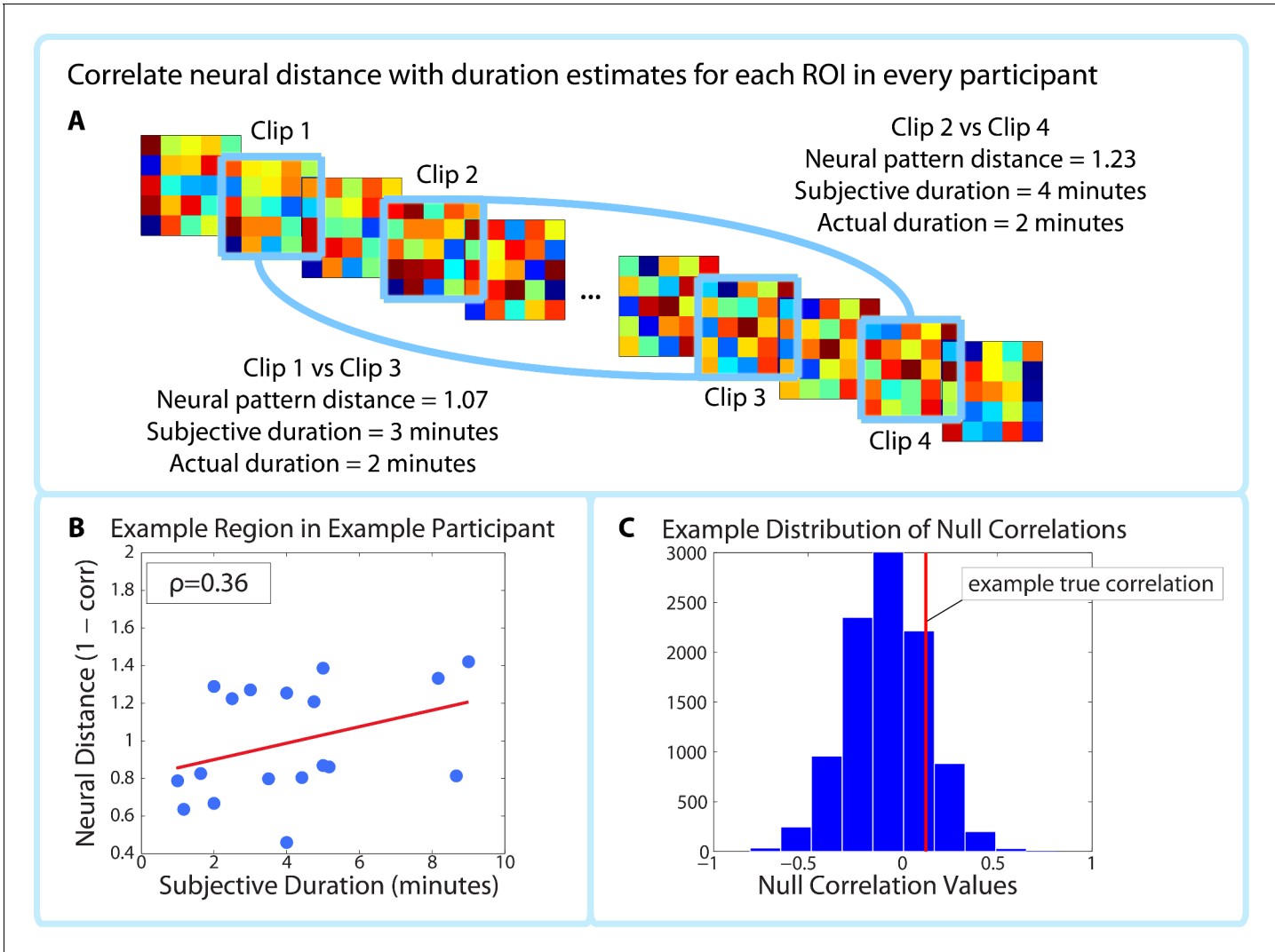

**Figure 4.** Correlating pattern distance with duration estimates within participants. For each ROI in each participant, the pattern distance between each pair of clips at encoding was correlated with the participant's retrospective duration estimate (A–B). The top panel (A) shows two example intervals. The neural distance (1-Pearson's *r*) between clips 2 and 4 (second interval) is greater than the neural distance between clips 1 and 3 (first interval), as is the subjective duration estimate. (B) shows the correlation between neural distance and duration estimates in a hypothetical region and participant. (C) We used a permutation test to generate 10,000 surrogate pattern distance vectors (see *Figure 4—figure supplement 1*), which we then used to obtain a distribution of null correlations between neural distances and duration estimates. For each ROI in each participant, we calculated the z-scored correlation value, which reflects the strength of the empirical correlation relative to the distribution of null correlations. For each ROI, we performed a random effects t-test to assess whether the z-score was reliably positive across participants. P-values from this t-test were then subjected to multiple comparisons correction using False Discovery Rate (FDR).

The following figure supplement is available for figure 4:

**Figure supplement 1.** Permutation test assessing the temporal specificity of correlations between pattern change and behavior.

In the within-participant analysis, we correlated each participant's duration estimates with that participant's neural pattern distances (see *Within-Participant Correlation between Pattern Change and Duration Estimates* and *Within-Participant Whole-brain Searchlight*). In the within-interval analysis, we correlated individual differences in subjective duration for a given interval with individual differences in neural pattern distance for that interval (see *Within-Interval Correlation between Pattern Change and Duration Estimates* and *Within-Interval Whole-brain Searchlight*). The two versions of each analysis were performed in order to rule out the possibility that our effects were driven either by participant or interval random effects. In particular, we were concerned that correlations between neural pattern distance and behavior could reflect sensitivity to perceptual or semantic features of the clips (i.e., clip pairs with similar perceptual/semantic features might be associated with shorter duration estimates and greater neural similarity, relative to clip pairs with more dissimilar features). The within-interval analysis addresses this concern by holding clip identity constant.

Next, we fit a mixed-effects model for each ROI (see *Mixed-Effects Model Accounting for Naïve Duration Estimates*), in which we estimated whether pattern distance in that ROI could predict duration estimates, even when accounting for participant random effects, item (interval) random effects, as well as naïve duration estimates (which are a proxy for the perceptual and semantic similarity between two clips, see *Behavioral results*).

Finally, we discuss the brain regions that showed significant effects across all analyses (see *Comparing Results from ROI and Searchlight Analyses*).

As noted in the Materials and methods, the ROI and searchlight analyses were conducted only on high-confidence two-minute intervals. Six-minute intervals were excluded from the fMRI analysis, since we could not successfully dissociate neural pattern change at this timescale from low-frequency scanner noise (see *Methodological challenges with analyzing pattern distance over long time scales* in the Materials and methods).

## Anatomical ROI analyses

We first tested whether pattern change in regions suggested by the literature to be important for representing temporal context (see *ROI Selection*) correlated with retrospective duration estimates. Anatomical ROIs were derived from FreeSurfer cortical parcellation (*Desikan et al., 2006*) and from a probabilistic MTL atlas (*Hindy and Turk-Browne, 2015*).

### Within-participant correlation between pattern change and duration estimates

The within-participant analysis procedure is outlined in *Figure 4*. We calculated the correlation between neural pattern distance and duration estimates within participants (*Figure 4A*) in each of the 32 ROIs shown in *Figure 5*. To assess the likelihood of obtaining a correlation of that magnitude by chance, we used a phase randomization procedure (described in Materials and methods) to obtain 10,000 null correlations for each ROI in every participant. This enabled us to calculate a Z-value for every ROI in every participant, which reflects the strength of the actual correlation between pattern distance and duration estimates relative to the distribution of null correlations (*Figure 4C*). Here we report the regions whose Z-values were consistently positive across participants, corrected for multiple comparisons using False Discovery Rate (*Benjamini et al., 2006*).

Out of the regions selected *a priori*, the right entorhinal cortex and right pars orbitalis showed a significant positive correlation between pattern change and duration estimates for high-confidence 2-minute intervals ($q<0.05$). *Figure 5* shows the mean Z-values across participants for all *a priori* ROIs (16 in each hemisphere), including lateral prefrontal regions (top panel A), medial temporal lobe regions, insula, putamen, and inferior parietal cortex (bottom panel B). While a large number of these regions had Z-values that were positive across participants (e.g., left hippocampus, left entorhinal cortex, right perirhinal cortex, right amygdala, bilateral insula, and right caudal middle frontal cortex, $p<0.05$ uncorrected), we report only those that survived FDR correction.

As part of an exploratory search, we also performed this analysis on the other brain regions derived from FreeSurfer cortical parcellation. This included the 16 ROIs mentioned above, in addition to regions in the occipital lobe, parietal lobe, medial prefrontal cortex, lateral temporal lobe, basal ganglia, thalamus and brainstem (the complete list of regions can be found in *Figure 5— source data 1*). Out of the 84 regions tested (42 in each hemisphere), the right entorhinal cortex, right pars orbitalis, and left caudal anterior cingulate cortex (ACC) showed significant positive

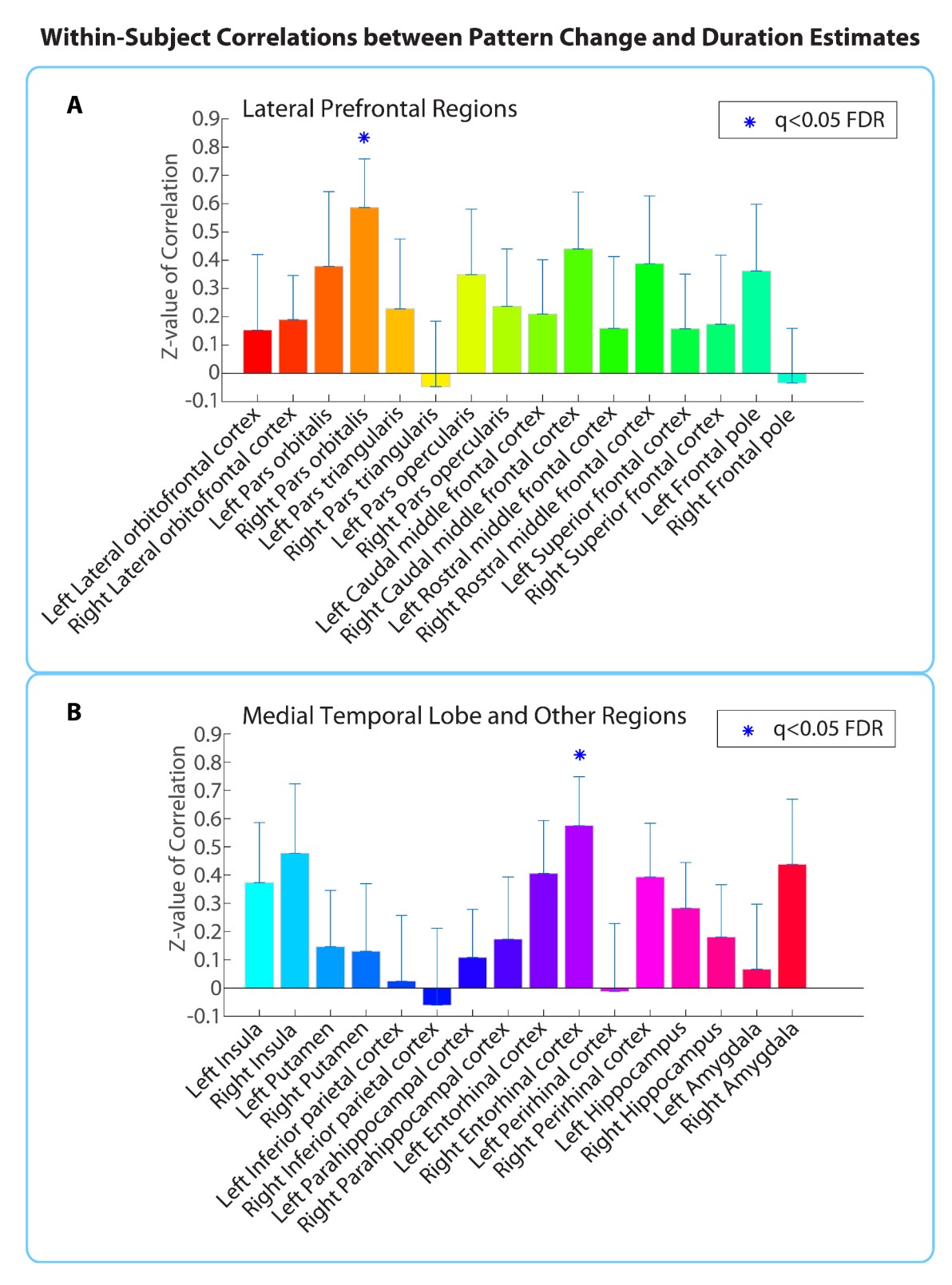

**Figure 5.** Within-participant ROI analysis: mean Z-values (across all 18 participants) of correlations between pattern distance and duration estimates for the 16 *a priori* ROIs. Z-values were obtained from the phase randomization procedure and reflect the strength of the empirical correlation relative to the distribution of null correlations. Error bars represent standard errors of the mean. The blue dots over the right entorhinal cortex and right pars orbitalis indicate that these ROIs survived FDR correction at q<0.05.

*Figure 5 continued on next page*

*Figure 5 continued*

The following source data and figure supplement are available for figure 5:

**Source data 1.** Within-participant analysis Z-values and Pearson's *r* values for all participants and grey matter regions derived from FreeSurfer segmentation and the probabilistic MTL atlas.

**Figure supplement 1.** Anatomical ROIs that showed a significant correlation between pattern change and duration estimates within participants, after whole-brain FDR correction.

correlations between pattern change and duration estimates ($q<0.1$). This suggests that the right entorhinal cortex and right pars orbitalis, which were part of our list of *a priori* ROIs, contained effects that were apparent even after whole-brain correction, and reveals an additional effect in the left caudal ACC that we had not anticipated. ***Figure 5—figure supplement 1*** displays the locations of these three regions in MNI space.

## Within-interval correlation between pattern change and duration estimates

Above, in the within-participants analysis, we found that the neural pattern distance between two clips at encoding was correlated with retrospective duration judgments in the right entorhinal cortex, right pars orbitalis and left caudal ACC. However, in the *Behavioral results*, we found that the perceptual and semantic similarity between two clips could explain some of the variance in subjective duration across intervals, even though it could not explain all the variance. Thus, it is possible that neural pattern change in the regions we found correlates with the component of duration estimates that is driven by perceptual and semantic content, rather than the component that is driven by abstract, slowly varying contextual features.

To rule out this concern, we performed a within-interval (across participants) version of the ROI analysis. For each ROI, we correlated (1) duration estimates for a given interval across participants with (2) the neural pattern distances for that interval across participants; results were then aggregated across all 2-min intervals. Rather than capturing variance within an individual across intervals of the story, this analysis captures variance across individuals for a given interval of the story. By performing the correlation within a given interval, we hold constant the perceptual and semantic content of the two clips and only leverage individual differences in how long the interval appeared retrospectively.

As described in the Materials and methods, a permutation test was used to assess the statistical significance of each correlation. Duration estimates were scrambled across participants 10,000 times to obtain a distribution of null correlations for every interval in every ROI. This enabled us to calculate a Z-value, which reflects the strength of the actual correlation between pattern distance and duration estimates relative to the distribution of null correlations. Finally, a right-tailed t-test was performed to assess whether the Z-values for a region were reliably above 0 across intervals. The p-values from this t-test were subjected to multiple comparisons correction using FDR.

Out of the regions selected *a priori*, the right entorhinal cortex, right amygdala, and right insula showed a significant positive correlation between pattern change and duration estimates for high-confidence 2-minute intervals ($q<0.05$). ***Figure 6*** shows the mean Z-values across intervals for all *a priori* ROIs (16 in each hemisphere).

Extending this analysis to the whole brain (same anatomical masks as in ***Figure 5—source data 1***) revealed only the right entorhinal cortex ($q<0.05$), suggesting that the effect in this region was strong enough to survive whole-brain correction.

Importantly, the right entorhinal cortex is the only region with significant effects in both the within-interval analysis (Cohen's $d = 0.83$) and the within-participant analysis (Cohen's $d = 0.79$). If neural pattern distance between two clips in entorhinal cortex were driven solely by changes in clip content, we would have expected the correlation with duration estimates to be larger for the within-participant analysis (where story content differed across intervals) than for the within-interval analysis (where story content is held constant). The fact that the effect sizes are similar shows that perceptual or semantic differences in content between the two clips are not the main factor driving the correlation between duration estimates and neural pattern change in this region.

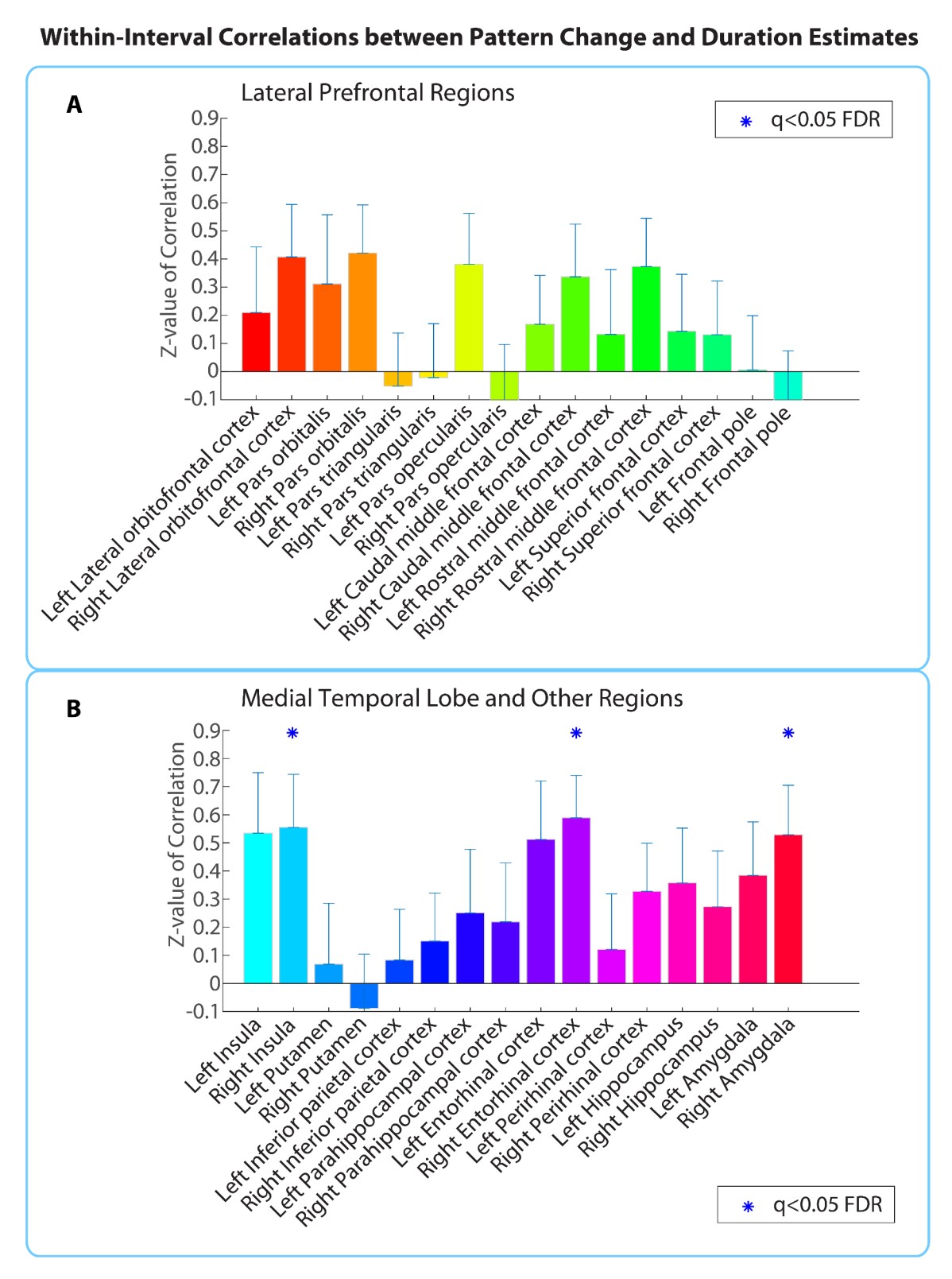

**Figure 6.** Within-interval ROI analysis: mean Z-values (across all 2-min intervals) of correlations between pattern distance and duration estimates for the 16 *a priori* ROIs. Error bars represent standard errors of the mean. Correlations between pattern change and duration estimates were performed across participants, separately for each interval.

*Figure 6 continued on next page*

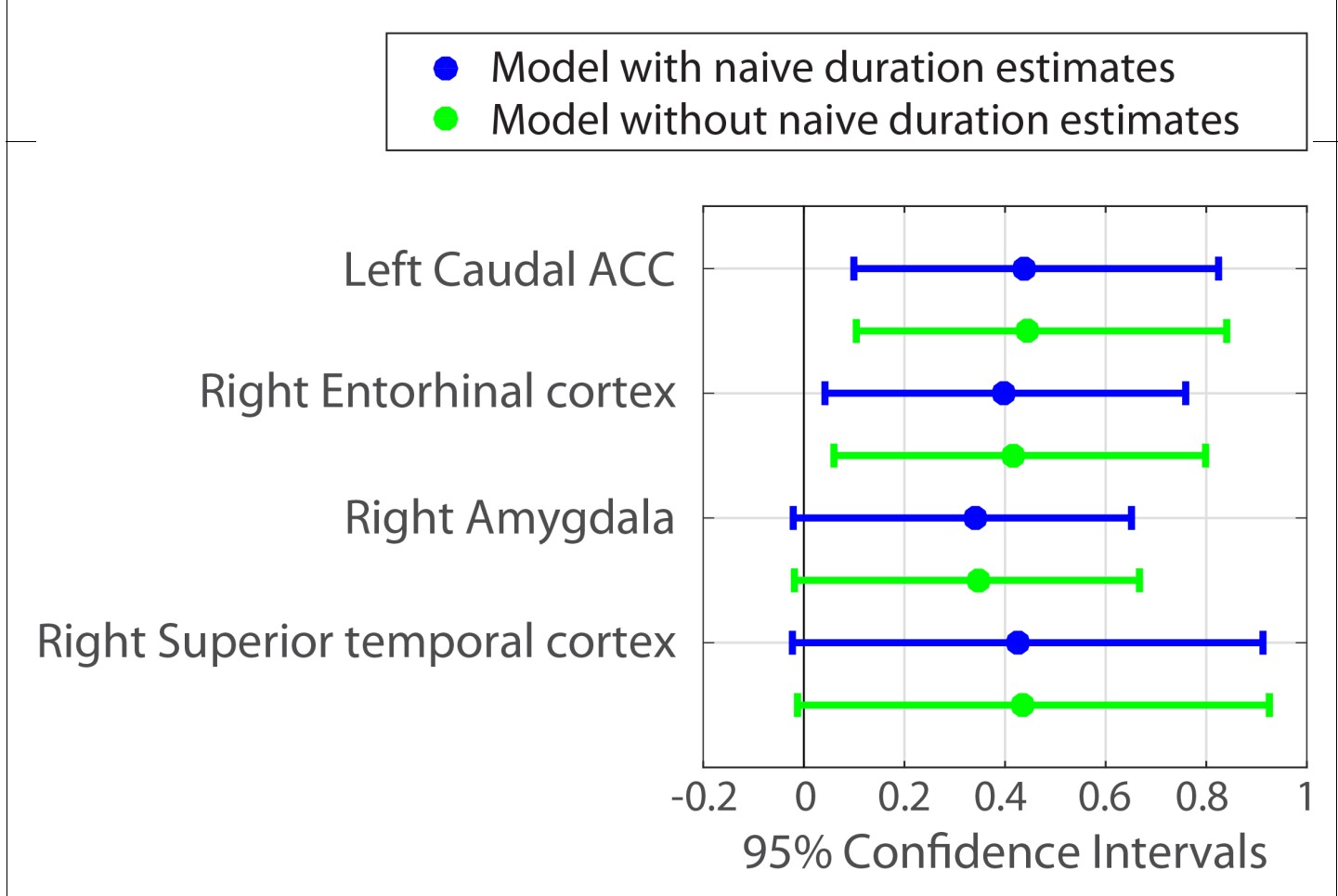

**Figure 7.** Parameter estimates and 95% confidence intervals for the fixed effect of neural pattern distance on duration estimates. We also included the right amygdala and right superior temporal cortex in the figure, because their confidence intervals did not include 0 when a slightly less conservative fitting procedure was used (see *Figure 7—source data 1* and Materials and methods).

The following source data is available for figure 7:

**Source data 1.** Parameter estimates (betas) and 95% confidence intervals for the fixed effects of neural pattern distance on duration estimates for all 84 anatomical regions.

## Mixed-effects model accounting for naïve duration estimates

We analyzed our data using a hierarchical linear regression model (*Gelman and Hill, 2006*; see Materials and methods for additional detail). This analysis estimates population-level effects of interest, while controlling for the possibility of individual variability between subjects and between clip pairs. In other words, this approach leverages the power of the within-interval analysis to control for the objective content similarity between two clips, while also taking into account variability in the effect across participants. In addition, we included the mean duration estimates from our naïve participants as a covariate in the model (see *Behavioral results*). Since naïve participants had estimated the temporal distance between each pair of clips without hearing the story, this covariate is a further control for the inherent guessability of the temporal distance between two clips. Both controls strengthen our interpretation that the remaining effect of neural pattern distance on duration estimates is driven by the contextual dissimilarity (rather than perceptual or content dissimilarity) between two clips.

For each anatomical region derived from FreeSurfer and MTL segmentation (42 in each hemisphere), we fit a model where duration estimates were predicted by naïve duration estimates as well

as the neural pattern distance in that region (see Materials and methods for the complete formula). We then computed 95% confidence intervals of the fixed-effects parameter estimates using the asymptotic Gaussian approximation (see Materials and methods).

The fixed effect of naïve estimates was positive in all models and its confidence intervals did not include zero in 80% of the models. This reproduced our finding that naïve duration estimates are correlated with the original duration estimates (see *Behavioral results*), suggesting that interval durations are partially guessable based on the similarity between clips. However, even under this control, the fixed effect of neural pattern distance in left caudal ACC and right entorhinal cortex exhibited confidence intervals that did not include zero (*Figure 7*). *Figure 7—source data 1* contains the parameter estimates and 95% confidence intervals for all 84 anatomical regions.

Importantly, including the naïve duration estimates as a covariate in the model did not significantly weaken the relationship between neural pattern distance and duration estimates in these regions (though the effects were slightly lower numerically). *Figure 7* shows in green the 95% confidence intervals for the same ROIs when naïve duration estimates are excluded from the model.

## Whole-brain searchlights

As with the Anatomical ROI analyses, both within-participant and within-interval analyses were performed for the Whole-Brain Searchlight analyses, in order to rule out the possibility that our effects were driven either by participant or interval random effects.

### Within-participant whole-brain searchlight

We ran a cubic searchlight with 3x3x3 (27) voxels (972 mm$^3$) through the entire brain and tested for a correlation between pattern change and duration estimates in each searchlight. The same phase-randomization procedure that was used for the within-participant anatomical ROI analysis was also applied here; this procedure generates Z-values that reflect how likely we are to get this strong of a correlation by chance, given the frequency spectrum of the fMRI data. When excluding low-confidence intervals, we found a significant cluster in the right anterior temporal lobe ($p=0.034$, FWE-corrected; Center of Gravity MNI coordinates (x, y, z) in mm: [45.6, −5.53, −21.7]; cluster size=572 voxels in 3 mm MNI space). Small parts of the cluster also extended to the right posterior insula and right putamen (see *Figure 8*).

### Within-interval Whole-brain searchlight

We also ran a searchlight version of the within-interval analysis. In order to match searchlights across participants, functional data were transformed to 3 mm MNI space. Since this transformation approximately doubles the number of brain voxels, we ran cubic searchlights of radius 2 with 5x5x5 (125) voxels through the entire brain.

As with the ROI analysis, this analysis was performed on high-confidence duration estimates. For each interval, we only included participants who had confidently recollected the temporal position of the two clips delimiting that interval.

To assess the significance of each correlation score, we used the same permutation test as for the ROI analysis. Duration estimates were scrambled across participants 10,000 times to obtain a distribution of null correlations, and Z-values were calculated for each interval. We thus obtained a brain map of Z-values for each of the 24 intervals, and FSL's *randomise* function was used to control the family-wise error rate, as above.

Similarly to the within-participant searchlight, we found a significant cluster in the right anterior temporal lobe ($p=0.019$, FWE-corrected; Center of Gravity MNI coordinates (x, y, z) in mm: [32.1, −10.2, −18.7]; cluster size=535 voxels in 3 mm MNI space). The cluster extended from the right parahippocampal gyrus, hippocampus and amygdala medially to the middle temporal gyrus and temporal pole laterally (see *Figure 9*).

## Comparing results from ROI and searchlight analyses

The within-participant ROI analysis revealed significant effects in the right entorhinal cortex, right pars orbitalis and left caudal ACC. The within-interval ROI analysis revealed significant effects in the right entorhinal cortex, right amygdala and right insula. The mixed-effects ROI analysis showed that the right entorhinal cortex and left caudal ACC had confidence intervals above 0, even when naïve

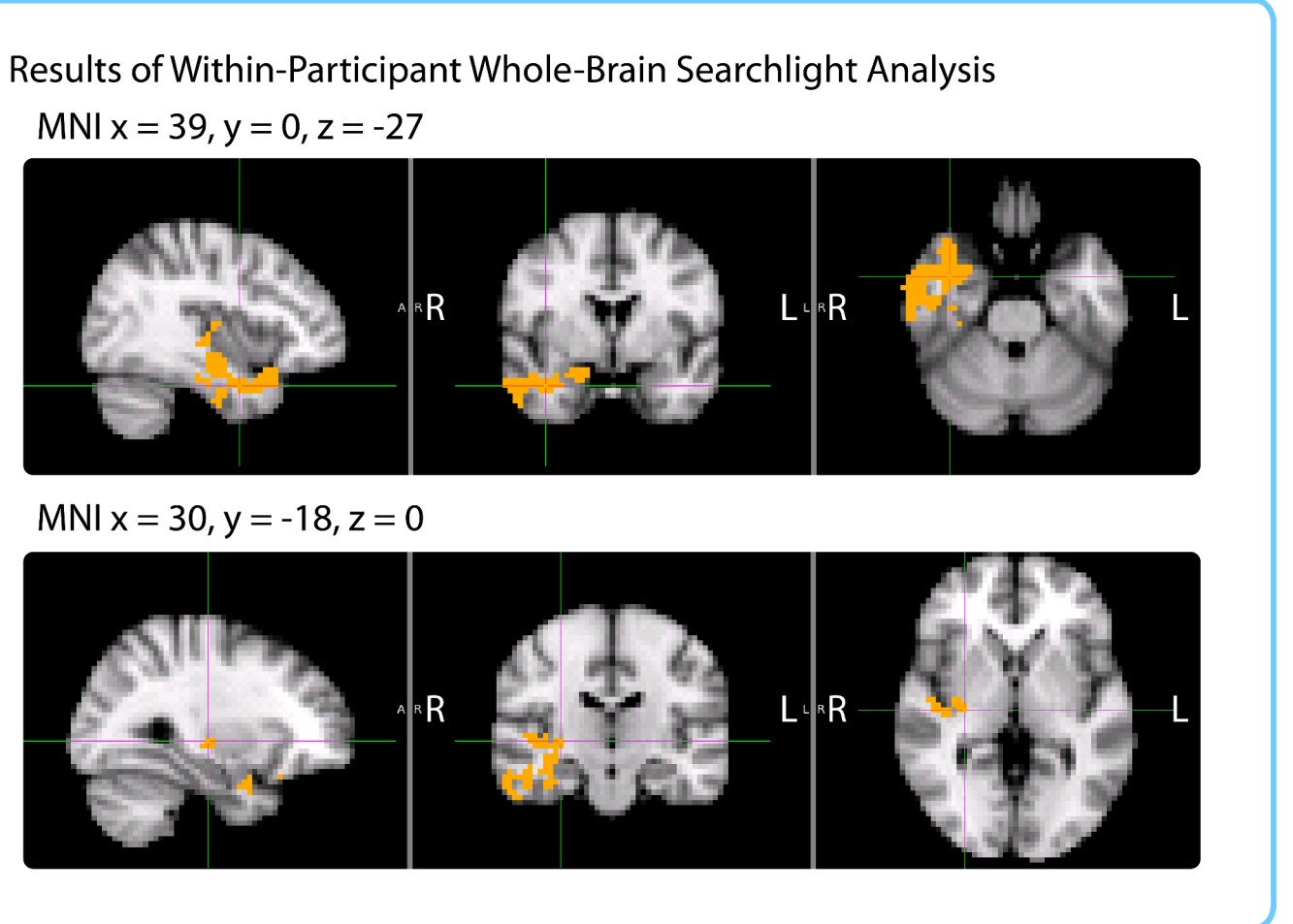

**Figure 8.** Results of within-participant whole-brain searchlight. Voxels in orange represent centers of searchlights that exhibited significant correlations between pattern change and duration estimates within participants across intervals ($p<0.05$, FWE). The significant cluster had peak MNI coordinates (in mm): $x = 45.6$, $y = -5.53$, $z = -21.7$.

duration estimates were accounted for. Both the within-participant and within-interval searchlights revealed significant clusters in the right anterior temporal lobe. *Figure 10* enables a comparison of the two searchlight analyses; the right entorhinal cortex ROI that emerged in all three ROI analyses is also overlaid. The within-interval searchlight cluster was located more medially than the within-participant searchlight cluster, though the two overlapped in the right amygdala, right temporal pole, and the cerebral white matter of the right anterior temporal lobe. Moreover, the within-interval searchlight cluster overlapped with the right entorhinal cortex ROI (see green voxels, *Figure 10*).

The difference in the set of regions that passed the significance threshold between the ROI and searchlight analyses is very likely due to the difference in shapes between the searchlight cube and the anatomical masks. Following the anatomy is particularly important for small, elongated regions like entorhinal cortex and caudal ACC, which are unlikely to be perfectly aligned across participants. For the searchlight analyses, the data needed to be transformed to MNI space in order to aggregate the results; consequently, imperfections in alignment can reduce the significance of searchlight results in these regions. On the other hand, anatomical ROI analyses were performed entirely in native space, making them more suitable for idiosyncratically shaped regions.

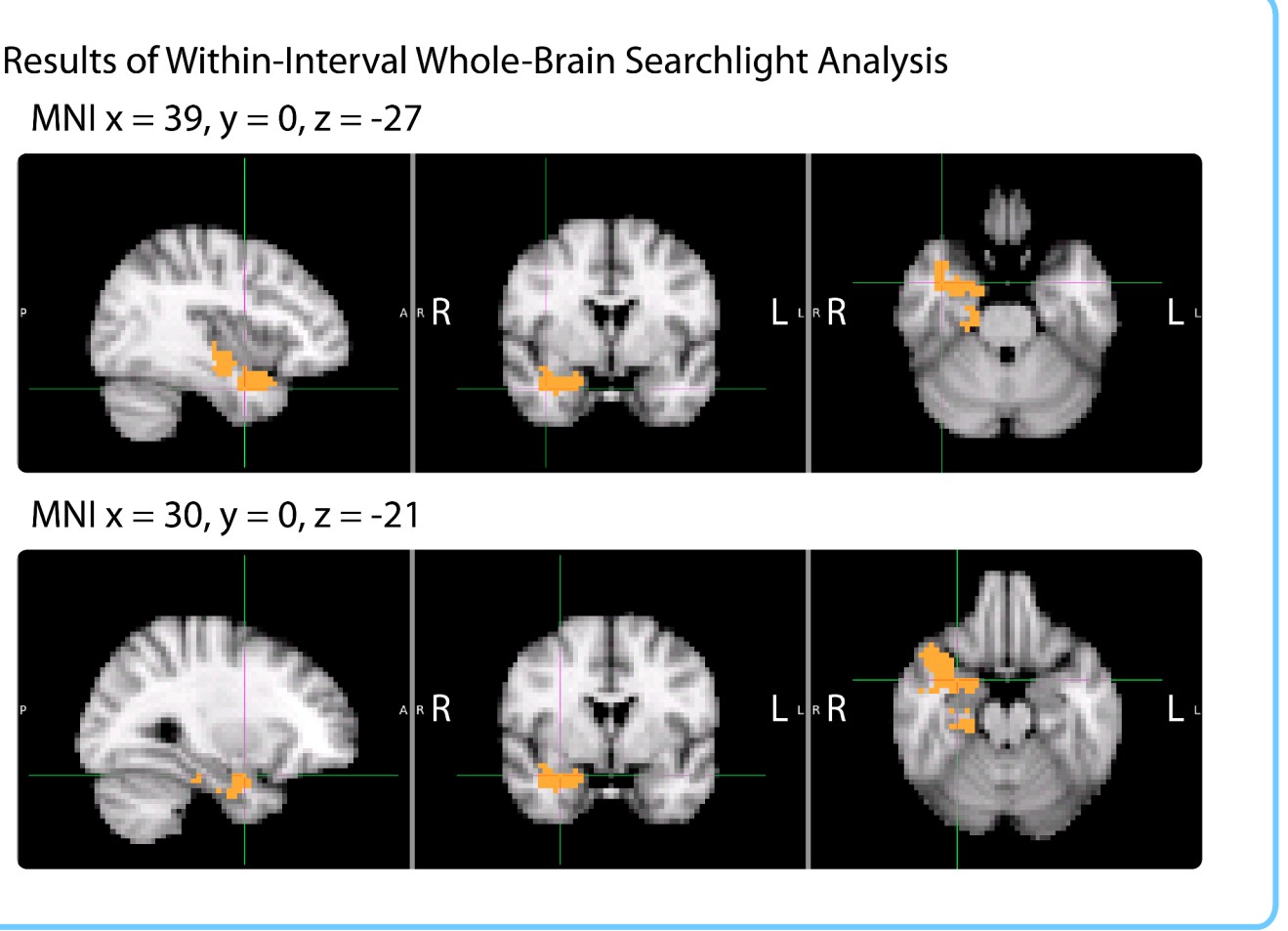

**Figure 9.** Results of within-interval whole-brain searchlight. Voxels in orange represent centers of searchlights that exhibited significant correlations between pattern change and duration estimates across participants ($p<0.05$, FWE). The significant cluster had center of gravity MNI coordinates (in mm): $x = 32.1$, $y = -10.2$, $z = -18.7$.

## Patterns of activity in entorhinal cortex change slowly over time

To further probe the idea that the regions we found represent slowly changing contextual features, we assessed whether their patterns of activity change slowly over time relative to the rest of the brain. We focused this analysis on the right entorhinal cortex and left caudal ACC, both of which were significant in the mixed-effects ROI analysis.

We quantified the speed of BOLD signal change in three different ways: (1) a multivariate procedure, (2) a multivariate procedure in which we regressed out ROI size, and (3) a univariate procedure. (1) For the multivariate procedure, we obtained the mean auto-correlation function of the pattern in every region, and took the full-width half-maximum (FWHM) of this function as a measure of how slowly the pattern moves away from itself on average (see Materials and methods). (2) Since this analysis was performed on anatomical masks derived from FreeSurfer parcellation, they varied substantially in size. To ensure that differences in the speed of pattern change were not due to differences in ROI size, we also performed the multivariate procedure after regressing the vector of ROI sizes (number of voxels) out of the vector of FWHM values for each participant. (3) Finally, we performed the above analysis for every voxel individually. Rather than calculating the mean auto-correlation function of the pattern in every region, we calculated the auto-correlation function of every voxel's time course and averaged the auto-correlation functions across all the voxels in a given

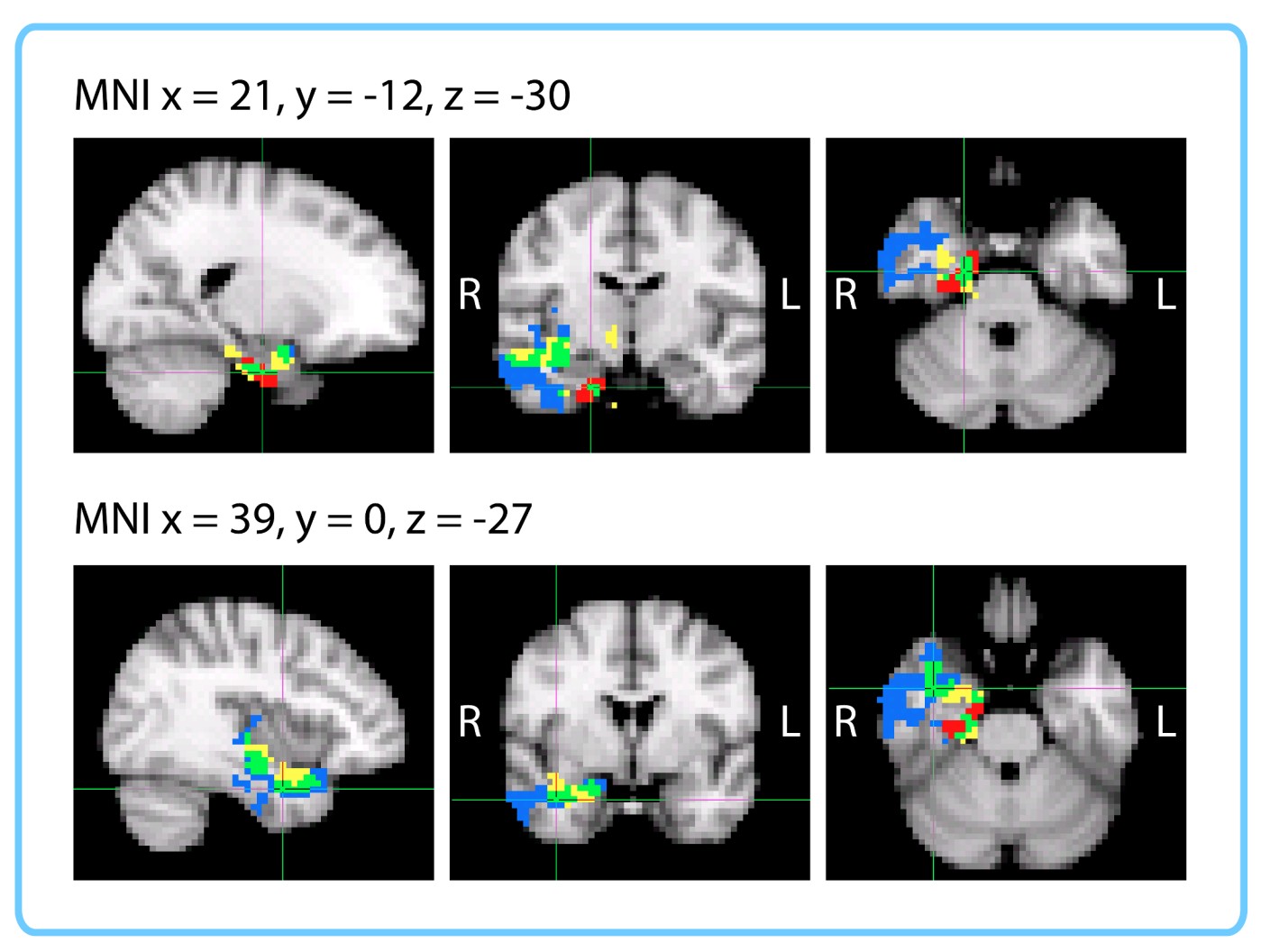

**Figure 10.** Comparison of ROI and Searchlight results. The within-participant searchlight cluster (*p*<0.05, FWE) is displayed in blue; the within-interval searchlight cluster (*p*<0.05, FWE) is displayed in yellow; voxels that overlap between the searchlights are shown in green. The right entorhinal cortex (*q*<0.05 FDR in both ROI analyses) is displayed in red; voxels that overlap between the within-interval searchlight and the right entorhinal ROI are shown in green.

region. The FWHM was then computed for this mean auto-correlation derived from individual voxel time courses.

Using these three procedures, we compared the FWHMs in the right entorhinal cortex and left caudal ACC with FWHMs in three regions known to be involved in auditory and language processing: the right transverse temporal cortex, which encompasses primary auditory cortex (*Destrieux et al., 2010*; *Shapleske et al., 1999*), the right banks of the superior temporal sulcus and the right superior temporal cortex, which are involved in auditory processing and the early cortical stages of speech perception (*Binder et al., 2000*; *Hickok and Poeppel, 2004*).

*Table 1* shows the FWHMs in the above regions derived using the three procedures, as well as the ranking of the right entorhinal cortex and left caudal ACC mean FWHMs relative to all the other masks in the brain (84 in total).

Across all three procedures, a right-tailed Wilcoxon signed-rank test indicated that the FWHMs in the right entorhinal cortex were consistently larger across participants than the FWHMs in the right transverse temporal cortex (*p*<0.00005, *p*<0.0005 and *p*<0.00005), the right banks of the superior temporal sulcus (*p*<0.001, *p*<0.001 and *p*<0.0005) and the right superior temporal cortex (*p*<0.005,

$p$=0.06 and $p$<0.0005). Thus, single voxels and multivariate patterns in entorhinal cortex changed consistently more slowly than those in regions involved in auditory and language processing. Moreover, the mean FWHM in the right entorhinal cortex was one of the largest among all 84 regions, ranking 3rd, 4th and 1st in the brain across the three procedures. The other regions with the slowest voxel and pattern change included the temporal pole, medial and lateral orbitofrontal cortex, frontal pole, perirhinal cortex, pars orbitalis and inferior temporal cortex.

On the other hand, the left caudal ACC ranked 66th, 67th and 46th out of 84 regions across the three procedures, suggesting that it did not exhibit slow signal change relative to the rest of the brain. Across the three procedures, the FWHMs in the left caudal ACC were larger than those in the right transverse temporal cortex ($p$<0.01, $p$<0.005, and $p$=0.059), but generally smaller than those in the right banks of the superior temporal sulcus ($p$=0.97, $p$=0.96, and $p$=0.42) and the right superior temporal cortex ($p$=1.0, $p$=1.0, and $p$=0.98). Thus, patterns in the left caudal ACC changed only slightly more slowly than those in primary auditory cortex.

Taken together, all three variants of the analysis showed that the right entorhinal cortex, along with other regions of the anterior and medial temporal lobe, orbitofrontal cortex and frontal pole, had the slowest pattern change in the brain. These results do not seem to be due to differences in the sizes of the anatomical masks and suggest that the right anterior MTL regions found most consistently in our ROI and searchlight analyses process information that changes slowly over time. Our findings are consistent with those of *Stephens et al. (2013)*, who showed that auditory cortex regions processing momentary stimulus features had intrinsically faster dynamics than higher-order regions that integrated information over longer time scales (see also *Lerner et al., 2011*).

## Story position effects cannot explain the correlation between duration estimates and neural pattern change

We found that duration estimates systematically decreased as a function of position in the story, with earlier intervals being estimated as longer than later intervals (*Figure 11*). The correlation between the estimated duration of an interval and its time in the story was consistently negative across participants ($M$=−0.40, $SD$= 0.22; $t$(16)=−7.59, $p$<0.00001).

This result may be a replication of the positive time-order effect: the finding that people judge earlier durations in a series of durations to be longer than later durations (*Block, 1982*, *1985*; *Brown and Stubbs, 1988*). The effect has been interpreted to mean that context usually changes more rapidly at the start of a novel episode (*Block, 1982*, *1986*). However, another possibility is that the characteristics of the particular story we picked are driving this result. In our story, there was a strong negative correlation between the mean number of event boundaries per interval and the position of the interval in the story ($r$=−0.77). Thus, the decrease in mean duration estimates with story position may be due to the relationship between the number of event boundaries and duration estimates (see *Behavioral results*).

If the decrease in duration estimates over time is due to a decrease in the amount of contextual change over the course of the story, we might expect BOLD pattern dissimilarity to decrease over time in the brain regions yielded by our ROI analyses. However, there was no consistent correlation between pattern change during an interval and its time in the story for the right entorhinal cortex

**Table 1.** Speed of pattern change in the right entorhinal cortex and left caudal ACC relative to the rest of the brain. Full-Width Half-Maximum (FWHM) values reflect how slowly patterns of activity (multivariate) or individual voxels (univariate) change over time. The Multivariate (-ROI size) column reflects the slowness of pattern change when controlling for the effect of ROI size.

|  | Multivariate | | Multivariate (-ROI size) | | Univariate | |
|---|---|---|---|---|---|---|
| Region | FWHM (TRs) | Ranking | FWHM (TRs) | Ranking | FWHM (TRs) | Ranking |
| Right entorhinal | $M$=18.9, $SD$=13.8 | 3rd | $M$=1.2, $SD$=1.9 | 4th | $M$=23, $SD$=15.6 | 1st |
| Left caudal ACC | $M$=8.3, $SD$=1.8 | 66th | $M$=-0.5, $SD$=0.5 | 67th | $M$=9.2, $SD$=3.8 | 46th |
| Right transverse temporal cortex | $M$=7.3, $SD$=1.2 | 80th | $M$=-0.8, $SD$=0.5 | 83rd | $M$=7.9, $SD$=1.2 | 68th |
| Right banks of superior temporal sulcus | $M$=9.0, $SD$=2.1 | 48th | $M$=-0.3, $SD$=0.4 | 49th | $M$=8.8, $SD$=1.7 | 61st |
| Right superior temporal cortex | $M$=11.0, $SD$=3.1 | 28th | $M$=0.4, $SD$=0.6 | 18th | $M$=10.3, $SD$=2.4 | 34th |

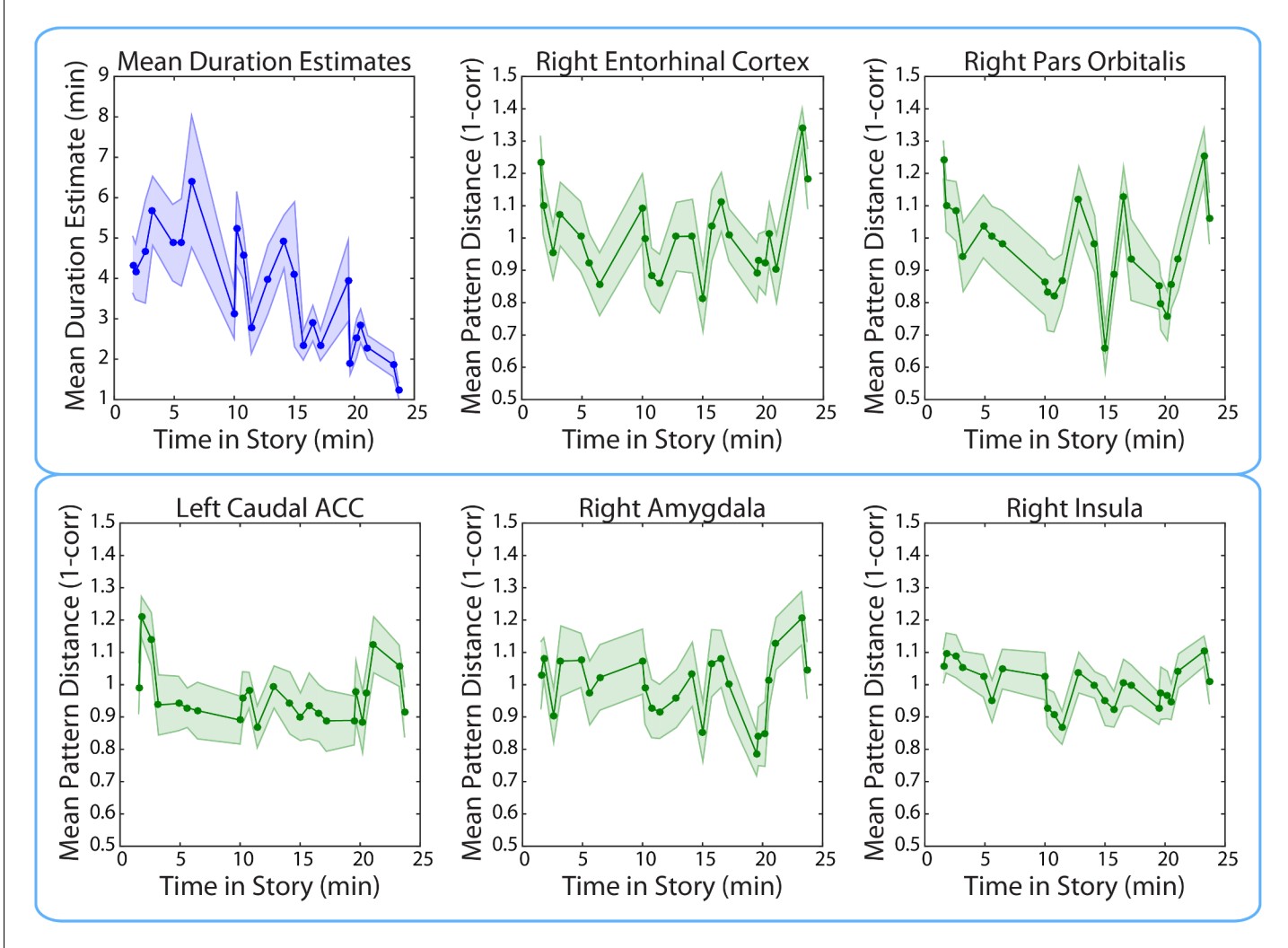

**Figure 11.** Mean duration estimates and pattern distances (across participants) for all 2-minute intervals as a function of the interval's position in the story. The middle time point of each 2-min interval (half-way between the two clips delimiting it) was chosen as the x-coordinate.

The following source data is available for figure 11:

**Source data 1.** Duration estimates and pattern distances in all FreeSurfer and MTL ROIs for each 2-minute interval in every participant.

($M$=0.03, $SD$=0.21; $t$(16)= 0.65; $p$=0.53), the right pars orbitalis ($M$=−0.10, $SD$=0.22; $t$(16)=−1.83, $p$=0.09), the left caudal ACC ($M$=−0.05, $SD$=0.18; $t$(16)=−1.15, $p$=0.27), the right amygdala ($M$=−0.02, $SD$=0.23; $t$(16)=−0.28, $p$=0.78) or the right insula ($M$=−0.08, $SD$=0.25; $t$(16)=−1.34, $p$=0.20). These results suggest that the relationship between duration estimates and pattern dissimilarity in these regions was not driven by a shared effect of story position. Rather, it seems that pattern dissimilarity in these regions correlated with more fine-grained variations in the estimated durations of nearby intervals (*Figure 11*).

To investigate why the above regions did not show the expected decrease in pattern dissimilarity over time, we assessed whether any brain region in the FreeSurfer or MTL atlas might show this effect. There was no brain region whose pattern of activity changed more at the beginning than at the end of the story. Given that we were looking for a slow change in neural signal (unfolding over the entire course of the story), we thought that our high-pass filter might be removing this slow change; to address this possibility, we analyzed the unfiltered data. When we did this, we found that neural pattern change in the unfiltered data showed a consistent correlation in the *opposite*

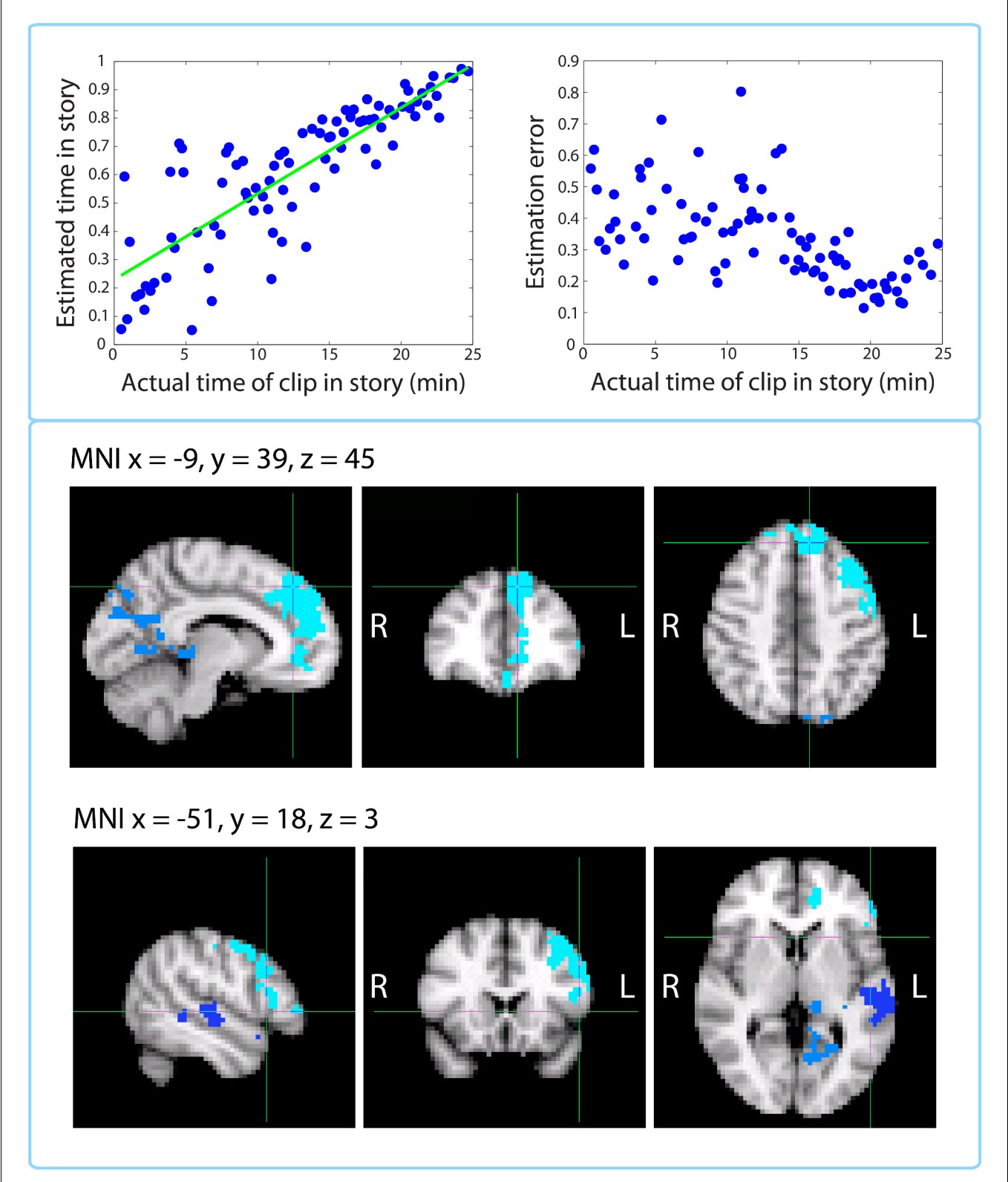

**Figure 12.** Replication of *Jenkins and Ranganath (2010)*: activity at encoding predicts accuracy of temporal context memory. **Top left panel**: Timeline estimates for a representative participant. The estimated temporal position of each clip is plotted against its actual position in the story. **Top right panel**: Group-averaged residual error for each clip plotted against its time in the story. Our behavioral results mimic those of Figure 2 in *Jenkins and Ranganath (2010)* showing that accuracy increases for clips that occurred later in the story. **Bottom panels**: Clusters that showed a significant
*Figure 12 continued on next page*

Figure 12 continued
correlation between activity at encoding and subsequent accuracy at placing a clip on the timeline of the story. The prefrontal cluster in light blue was significant (*p*=0.008, FWE), while the medial parietal cluster (*p*=0.058, FWE) and the lateral temporal cluster in dark blue (*p*=0.098, FWE) were trending.

direction: almost all brain patterns changed more at the end of the story than at the beginning, including the CSF and white matter ($q < 0.05$, FDR), suggesting that a signal unrelated to neural processing, such as scanner drift or motion, may cause activity patterns to change more as time passes (see *Figure 11—source data 1*). Thus, even if the degree of neural pattern change were decreasing over time, we might not be able to detect this effect, as it would have to overcome a global signal in the opposite direction that is not due to neural activity and that is present everywhere, including the CSF.

## Replication of *Jenkins and Ranganath (2010)*: activity at encoding predicts accuracy of temporal context memory

As described in the Materials and methods (*Time perception test* section), besides estimating the elapsed duration between pairs of clips from the story, participants were given an additional test, where they indicated each clip's position on the timeline of the story. The mean correlation (across participants) between the actual and estimated temporal position on the timeline of the story was $r = 0.885$ (SD=0.05), suggesting that participants remembered the temporal position of each clip extremely well ($p < 10^{-21}$). *Figure 12* shows the timeline estimates for a representative participant (top left panel), as well as the absolute residual error associated with each clip (top right panel), group averaged and plotted against time in the story.

This behavioral dataset enabled us reproduce an fMRI analysis from *Jenkins and Ranganath (2010)*, where voxel activity at encoding was correlated with subsequent accuracy in remembering when a trial occurred in the experiment. For each participant, we regressed the estimated timeline position against the actual position and used the absolute value of the residual as a measure of error. We found that the accuracy (negative error) of timeline placements was significantly correlated with encoding activity in large clusters of the left dorsolateral prefrontal cortex and medial prefrontal cortex, including dorsomedial PFC and anterior cingulate ($p = 0.008$, FWE-corrected; Center of Gravity MNI coordinates (x, y, z) in mm: [−20, 34.8, 28.4]; cluster size = 1121 voxels in 3 mm MNI space), as well as sub-threshold clusters in the medial parietal cortex, including precuneus and posterior cingulate ($p = 0.058$, FWE-corrected; Center of Gravity MNI coordinates (x, y, z) in mm: [−10.5, −54, 16.1]; cluster size = 419 voxels), and left superior temporal gyrus ($p = 0.098$, FWE-corrected; Center of Gravity MNI coordinates (x, y, z) in mm: [−56.9, −19.1, −3.72]; cluster size = 270 voxels).

## Discussion

While human and animal time perception has been a subject of intense empirical investigation (see *Wittmann, 2013*), most neuroimaging studies have tested its mechanisms on the scale of milliseconds to seconds and neglected paradigms in which long-term memory plays an important role. Such studies have typically employed *prospective paradigms*, in which participants must deliberately attend to the duration of a stimulus. However, behavioral studies in humans have consistently demonstrated that *retrospective paradigms*, in which participants are asked to estimate the duration of an elapsed interval from memory, tap into different cognitive mechanisms from prospective ones (*Hicks et al., 1976*; *Zakay and Block, 2004*; *Block and Zakay, 2008*). In retrospective paradigms, changes in spatial, emotional and cognitive context tend to modulate estimates of elapsed time (*Block, 1992*; *Block and Reed, 1978*; *Sahakyan and Smith, 2014*; *Pollatos et al., 2014*).

In the present study, we used changes in patterns of BOLD activity as a proxy for mental context change. We sought to extend previous neuroimaging work by testing whether neural pattern change predicts duration estimates on the scale of several minutes and in a more naturalistic setting, where spatial location, situational inference, characters, and emotional elements can all drive contextual change.

Participants were scanned while they listened to a 25-minute radio story and were subsequently asked how much time (in minutes and seconds) had elapsed between pairs of clips from the story (all pairs were in fact two minutes apart). Using this approach, we were able to probe retrospective duration memory repeatedly within participants without needing to interrupt the encoding of the story. This allowed us to leverage within-participant variability in neural pattern change and relate it to a participant's retrospective duration estimates.

Using a within-participant anatomical ROI analysis (encompassing 16 regions selected *a priori*), we found that neural pattern distance in the right entorhinal cortex and right pars orbitalis at the time of encoding was correlated with subsequent duration estimates. Extending this analysis to all anatomical ROIs in cortex revealed an additional effect in the left caudal anterior cingulate cortex (ACC). These results converged qualitatively with the results of our whole-brain searchlight analysis, which revealed a significant cluster spanning the right anterior temporal lobe.

To test our interpretation that duration estimates were driven by contextual change, we asked a separate group of participants to identify event boundaries in the story. We found that the number of event boundaries between two clips was very highly correlated with participants' subsequent duration estimates. Importantly, the number of event boundaries was significantly less correlated with duration estimates for a separate group of 'naïve' participants, who had been asked to estimate the elapsed time between clips without first hearing the story. These behavioral experiments provide evidence that retrospective duration estimates were indeed influenced by memory for intervening contextual changes between clips.

In addition, we sought to rule out the possibility that neural pattern distance between two clips reflected only the perceptual or semantic similarity between them, rather than the degree of mental context change. We performed a within-interval analysis, in which pattern distances for the same pair of clips were correlated with duration estimates *across participants*. The within-interval ROI analysis yielded effects of the same size in the right entorhinal cortex, right amygdala and right insula. The within-interval whole-brain searchlight revealed a significant cluster in the right anterior temporal lobe. Thus, pattern distance in the right anterior temporal lobe, particularly the right entorhinal cortex, predicted variability in duration estimates even when the perceptual and semantic distance of the clips was controlled as much as possible, suggesting that pattern change in these regions may capture idiosyncratic differences in mental context that cannot be predicted from the stimulus alone.

Finally, if neural pattern distance between two clips reflected only the similarity in content between them, rather than abstract contextual similarity, we would expect the correlation between pattern distance and duration estimates to be weakened when controlling for naïve duration estimates, which were based solely on the perceptual and semantic similarity between two clips. Fitting a mixed-effects model to each ROI showed that neural pattern distance in the right entorhinal cortex, along with the left caudal ACC, exhibited a significant effect on duration estimates even when all other factors, including random effects of participants and intervals, as well as naïve duration estimates, were controlled for.

In support of the hypothesis that these regions represent slowly varying contextual information, we found that the right entorhinal cortex, as well as adjacent regions of the MTL, temporal pole and orbitofrontal cortex, had some of the slowest neural pattern change in the entire brain. This is in line with findings that brain regions at the top of the processing hierarchy (furthest from the primary perceptual areas) integrate information over longer time scales and are therefore best suited for representing abstract information extracted from multiple streams of sensory observations (*Stephens et al., 2013*; *Lerner et al., 2011*).

Our results implicating the right entorhinal cortex in representing context fit well with other results in the literature. Multiple lines of evidence have suggested an important role for the entorhinal cortex in representing relationships between the spatial environment, task and incoming stimuli. Lesions of the lateral entorhinal cortex in rodents have shown that this region is necessary for discriminating between novel and familiar associations of object and place, object and non-spatial context, or place and context, while leaving non-associative forms of memory unaffected (*Buckmaster et al., 2004*; *Wilson et al., 2013a*; *2013b*). Moreover, electrophysiological recordings in rats performing a spatial memory task showed that neurons in the medial entorhinal cortex exhibited greater context sensitivity and greater modulation by task-relevant mnemonic information than hippocampal neurons, while hippocampal neurons carried more specific spatial information (*Lipton et al., 2007*). Medial entorhinal neurons also exhibited longer firing periods, which led the

authors to propose that they could bind a series of hippocampal representations of distinct events (*Lipton and Eichenbaum, 2008*). Thus, changes in distributed entorhinal activity patterns on the scale of minutes might represent changes in contextual elements that are later retrieved to make duration judgments (for theoretical discussion of the role of entorhinal cortex in contextual representation, see *Howard et al., 2005*).

While the right entorhinal cortex was the only medial temporal lobe region that survived FDR correction in both our within-participant and within-interval ROI analyses, our whole-brain searchlights found a significant relationship between pattern change and duration estimates in two extensive clusters that overlapped in the right hippocampus, the right perirhinal cortex, right amygdala and right temporal pole.

Two previous studies, *Noulhiane et al. (2007)* and *Ezzyat and Davachi (2014)*, have directly implicated the MTL in retrospective time estimation in humans. *Ezzyat and Davachi (2014)* scanned participants while they were presented with trial-unique faces and objects on a scene background, which changed every four trials. After each run, participants were asked whether pairs of stimuli had occurred close together or far apart in time (all pairs were about 50 s apart). They found that neural pattern distance in the left hippocampus at the time of encoding was greater for pairs of stimuli later rated as 'far apart', though only when the stimuli were separated by a scene change. *Noulhiane et al. (2007)* used a retrospective behavioral paradigm similar to ours in patients with unilateral MTL lesions. In that study, participants were asked to estimate the temporal distance between object pictures that had been randomly inserted into a silent documentary film. They found that the degree of left entorhinal, left perirhinal and left temporopolar cortex damage correlated with the degree to which patients overestimated minutes-long intervals in retrospect. (For related evidence from the animal literature, see *Jacobs et al., 2013*, who showed that bilateral inactivation of the hippocampus impaired rats' ability to discriminate between similarly long durations, such as 8 and 12 minutes, but not between less similar intervals, such as 3 and 12 minutes.)

Our ROI and searchlight results are in line with the above set of findings, and suggest that patients with anterior MTL lesions might be impaired in retrospective time estimation because patterns of activity in entorhinal, perirhinal, and temporopolar cortex encode contextual changes on the scale of minutes. The set of regions we found is more extensive than those in *Ezzyat and Davachi (2014)* and mostly right-lateralized. It is possible that the difference in the extent of our effects could be explained by differences in the paradigms that were used. In both the *Noulhiane et al. (2007)* and *Ezzyat and Davachi (2014)* studies, the links between objects and their context had to be deliberately constructed. In our study, the clips whose temporal distance participants estimated were excerpts from a story, and therefore strongly linked with a situational, spatial, and emotional context. Thus, it is possible that activity patterns in a more extensive cluster tracked temporal distance estimates because our auditory story caused changes in a broader set of contextual features.

Extending our anatomical ROI analysis to the entire brain showed that pattern change in the left caudal anterior cingulate cortex (ACC) predicted subsequent duration estimates, and this region remained significant in a mixed-effects model controlling for the effect of naïve duration estimates. However, caudal ACC exhibited more rapid pattern change than the anterior and medial temporal lobe, suggesting that it may represent a qualitatively different, faster-changing signal. Caudal ACC activity has been shown to increase in response to shifts in task contingencies (see *Shenhav et al., 2013*, for a review) and there is converging evidence that ACC responses are important for adjusting behavior to unexpected changes by increasing attention and learning rate (*Bryden et al., 2011*; *Behrens et al., 2007*; *McGuire et al., 2014*). *O'Reilly et al. (2013)* have provided evidence that the ACC only responds to surprising outcomes when they necessitate updating beliefs about the current state of the world. Although the present study was not designed to test such accounts, our findings are consistent with a role for ACC in updating predictive models. Events in the story that prompt participants to update their beliefs about the characters' situation are also likely to cause changes in cognitive context and therefore overestimation of duration. However, future studies are needed to test this interpretation, for instance by manipulating belief updating independently of surprise and measuring its effect on retrospective duration estimates.

In addition to the anatomical ROI analysis, we performed a whole-brain searchlight that yielded an extensive cluster covering the right anterior temporal lobe, extending from the medial temporal regions described above to the middle temporal gyrus and temporal pole. Prior work has suggested that the middle temporal gyrus and temporal pole are involved in narrative comprehension

(*Ferstl et al., 2008*; *Mar, 2004*) and narrative item memory (*Hasson et al., 2007*; *Maguire et al., 1999*). *Ezzyat and Davachi (2011)* found a similarly located cluster (extending from the right perirhinal cortex to the right middle temporal gyrus) to be involved in integrating information within narrative events. In particular, they showed that activity within these regions gradually increases within events and that this increase predicts the degree to which memories become clustered within events. Retrospective time judgments have been shown to increase with the number of events an interval contains (*Poynter, 1983*; *Zakay et al., 1994*; *Faber and Gennari, 2015*), suggesting that brain regions involved in clustering memories by events may carry important information for estimating durations.

Finally, we were able to replicate an analysis by *Jenkins and Ranganath (2010)*, who showed that activity during encoding in the left lateral prefrontal cortex and right anterior hippocampus predicted accuracy in remembering when a trial had occurred in the experiment. Our analysis revealed a cluster in the left dorsolateral prefrontal cortex that is similar to that found in their study. However, we also found significant clusters in the medial prefrontal and medial parietal cortex. These regions may be important for maintaining narrative information over minutes-long timescales (*Lerner et al., 2011*; *Hasson et al., 2015*; *Chen et al., 2015*), which might explain why their activity predicted temporal context memory for clips from an auditory story, but did not appear in *Jenkins and Ranganath (2010)*, where participants recalled the timing of trials which were not linked by a narrative. Moreover, our clusters overlap highly with the 'posterior medial network' (*Ritchey and Ranganath, 2012*), which has been consistently implicated in episodic memory, episodic simulation and theory of mind.

## Conclusion

After probing human participants' time perception for intervals from an auditory story they had just heard, we found substantial variability in subjective estimates of the passage of time. This variability was significantly correlated with changes in BOLD activity patterns in the right anterior temporal lobe, particularly the right entorhinal cortex, between the start and end of each interval. Control experiments demonstrated that duration estimates were strongly driven by contextual boundaries and that the relationship between neural distance and behavior still held when we controlled for the perceptual and semantic similarity of the clips. Our findings suggest that patterns of activity in these regions might encode contextual information that participants can later retrieve to infer the durations of intervals on the scale of minutes. Additional work is needed to assess how these regions contribute to representing particular contextual features (such as physical environment, abstract task states, and emotional states) and whether changes in each of these features affect retrospective duration estimates differently.

## Materials and methods

### Participants

18 participants (13 female) took part in the study. All participants were recruited from the Princeton undergraduate and graduate student population and were between 18 and 31 years of age (mean = 22 years). All participants were screened to ensure no neurological or psychiatric disorders. Written informed consent was obtained for all participants in accordance with the Princeton Institutional Review Board regulations. Participants were compensated $20/hr for the scanning session, and $12/hr for the behavioral session.

Given that no previous studies had related neural pattern change during a naturalistic stimulus to subsequent duration estimates for minutes-long intervals, we could not *a priori* estimate the variance in the pattern change signal, the variance in duration estimates, or the correlation between them. Therefore, rather than performing a power analysis, we chose a sample size that was in the same range as previous fMRI studies that had used naturalistic stimuli to study memory (*Lerner et al., 2011*, n=11 per condition; *Chen et al., 2015*, n=13, 14 and 24 per condition; *Chen et al., 2016*, n=22 [5 excluded]), as well as fMRI studies that had related neural pattern distance to mnemonic judgments (*Ezzyat and Davachi, 2011*, n=19; *Jenkins and Ranganath, 2010*, n=16 (1 excluded); *Ezzyat and Davachi, 2014*, n=21 (3 excluded), *Jenkins and Ranganath, 2016*, n=17).

## Experimental design and stimuli

The experiment consisted of two parts: an approximately 40-min session in the MRI scanner, during which participants listened to the auditory story, followed immediately by a 1-hr behavioral session, during which participants completed a time perception test on the story they had just heard. *Figure 1* illustrates the experimental procedure.

### fMRI session

Prior to the fMRI session, participants were instructed to listen carefully to the auditory story while in the scanner, because they might be asked questions about it later. The nature of the follow-up questions was unknown to the participants. While in the scanner, participants listened to a 25-minute-long radio adaptation of a science fiction story called 'Tunnel Under the World' (written by Frederik Pohl), originally aired on the radio drama series, 'X Minus One', in 1956.

### Time perception test

After leaving the scanner, participants were surprised with a time perception test, presented on a laptop with the Psychophysics toolbox (*Brainard, 1997*; *Pelli, 1997*) for MATLAB (The MathWorks Inc., Natick, MA). For each of 43 questions, participants listened to a 10 s clip from the story, followed by another 10 s clip, and were asked to estimate how much time had passed between the first and second clips when they initially heard the story. Participants were specifically asked to estimate how much time had passed in their own lives, rather than how much narrative time had passed in the story. They were also asked to make the judgments as intuitively as possible, without resorting to deductive reasoning about the sequence of events that unfolded in between the two excerpts.

Participants had complete control over the pacing of the test. On each question, they initiated the playing of the clips, and were able to replay the clips if they missed them the first time. They could take as long as they wished to enter their duration estimates (in minutes and seconds), using the keyboard. Clip pairs were identical across participants, but the order in which the pairs were presented was randomized.

To control for the objective passage of time, we ensured that 24 of the clip pairs were 2 minutes apart and 19 of the pairs were 6 minutes apart. Debriefing showed that participants were unaware of this manipulation, and the high variability of duration estimates for both the 2 and 6-min intervals further confirmed that they were unaware of the fixed interval durations.

After participants had provided duration estimates for all 43 intervals, the 86 clips that had delimited those intervals were replayed in a random order (unpaired), and participants were asked to place each clip on the timeline of the story. For each of the 86 questions, a white line appeared on a black background, representing the full length of the story. Participants could place their cursor at any point on that line, followed by the Enter key. After each placement, they were asked to provide a confidence rating on a scale of 1 to 5, reflecting their confidence about that clip's place in the story. Participants were instructed to base the confidence rating on their certainty of when that clip occurred in the story, rather than on the vividness of the memory for that clip.

Please note: the first of our 18 participants completed a version of the time perception test that differed only in the following way: the specific intervals in the story whose duration was asked about were different. In all other respects (half of the intervals were 2 min while the other half were 6 min apart), the behavioral test was identical to the subsequent 17 participants. For this reason, however, any analyses where duration estimates are compared across participants were performed on 17 rather than 18 participants. Any within-participant analyses were performed on all 18 data sets.

### Naïve time perception test

To address the concern that participants were estimating temporal distance between two clips based purely on the content of the clips (rather than their memory of when the clips had occurred in the story), we administered an identical time perception test to a separate group of 17 participants who had never heard the story. Naïve participants were asked to try their best to guess how much time passed between each pair of clips during the original telling of the story, even though they had never heard the story. Participants were told the length of the story (25 min, 33 s) and informed that the maximum distance between two clips could not exceed that duration.

### Event boundary test

A separate group of 9 participants were asked to listen to the same story and to press the space bar every time they thought an event had ended and a new event was beginning. This test was purely behavioral and fMRI data were not collected for these participants.

## Behavioral data analysis

### Significance of correlation between duration estimates and event boundaries

To assess whether the number of event boundaries in an interval predicted duration estimates for that interval, we related our original participants' duration estimates with event boundary data collected from a separate group of 9 participants. For each 2-min interval from the time perception test, we counted the number of event boundaries that a participant had indicated during that interval and averaged that number across the 9 participants. This resulted in a mean number of event boundaries per interval, which was then correlated with the mean estimated duration of that interval from our original participants.

To assess the statistical significance of this correlation, we performed a bootstrapping procedure on the duration estimates. We obtained 1000 bootstrap samples, each time selecting with replacement a different subset of $n$ individuals from our pool of $n$ participants. The duration estimates for each subset were averaged across participants and correlated with the mean number of event boundaries. The upper limit ($ul$) for an $x$% confidence interval was set to the value of the Pearson correlation in percentile $x$% of the bootstrap distribution; the lower limit ($ll$) for the confidence interval was set to the value of the Pearson correlation in percentile $100-x$ of this distribution. Confidence intervals that did not encompass zero were considered reliable at the given level of confidence.

### Significance of difference in correlations with event boundaries between original duration estimates and naïve duration estimates

We hypothesized that duration estimates from our original participants (who had actually heard the story) would be significantly more correlated with the number of event boundaries between two clips than duration estimates from our naïve participants, who had never heard the story. To assess the significance of the difference in correlations, we computed the $r_{diff}$ (empirical difference), as well as the upper confidence limits ($ul_{diff}$) and lower confidence limits ($ll_{diff}$) for the difference between the two correlations. We used the following formulae (*Zou, 2007*; *Poppenk and Norman, 2012*) for two bootstrapped correlation confidence intervals:

$$r_{diff} = r_1 - r_2$$

$$ll_{diff} = r_1 - r_2 - \sqrt{(r_1 - ll_1)^2 + (ul_2 - r_2)^2}$$

$$ul_{diff} = r_1 - r_2 + \sqrt{(ul_1 - r_1)^2 + (r_2 - ll_2)^2}$$

The upper ($ul_1, ul_2$) and lower limits ($ll_1, ll_2$) for a 95% confidence interval of each group's correlation were calculated as described above.

### Reliability of duration estimates across participants within and between groups

We hypothesized that both our original participants and the naïve participants (who had never heard the story) would use consistent strategies to estimate the temporal distance between two clips, but that these strategies would differ across groups. If this is the case, duration estimates should be more reliable across participants within groups than across participants between groups.

To assess within-group reliability, we correlated each participant's duration estimates with the mean of the other participants' estimates. These correlations were then averaged across participants within a group to obtain a mean within-group ISC (inter-subject correlation). The between-group reliability was calculated by correlating each participant's duration estimates from one group (e.g., the

original participants) with the mean duration estimates from the other group (e.g., the naïve participants). These correlations were then also averaged across participants to obtain a mean between-group ISC. Confidence intervals for the mean between-group ISC were calculated by bootstrapping the duration estimates from both groups 10,000 times, each time selecting with replacement a different subset of $n$ individuals from our pool of $n$ participants. The between-group ISCs were calculated for each bootstrap sample and averaged across participants, resulting in a distribution of 10,000 mean between-group ISCs. Confidence intervals for the within-group ISC were obtained in a similar manner.

To assess the significance of the difference between the mean within-group ISC and the mean between-group ISC, we compared the empirical difference with a null distribution of differences. Group labels (naïve participants vs. original participants) were scrambled 10,000 times, such that each participant's duration estimates were randomly assigned to either the naïve group or to the original group. The difference between the mean within-group ISC and the mean between-group ISC was then computed for these two random groups. Using this null distribution of ISC differences, we calculated a p-value based on the number of permutations that yielded a greater difference than the empirical difference.

Please note that the within-group and between-group correlations could be compared only because the group sizes were identical (17 participants in each) and because the within-group correlations were equally strong for the original and naïve groups ($M$=0.43, $SD$=0.25, 95% CI=[0.37, 0.58] vs. $M$=0.43, $SD$=0.18, 95% CI [0.40, 0.56]). Since the within-group ISCs are comparable, we can infer that the significant difference between the within-group and between-group reliability reflects a difference in the signals (strategies) underlying the two groups of duration estimates (*Chow et al., 2015*), rather than a difference in within-group reliability.

## MRI acquisition

Participants were scanned in a 3T full-body Skyra MRI scanner (Siemens, Munich, Germany) with a 20-channel head coil. Functional images were acquired using a T2*-weighted echo planer imaging (EPI) pulse sequence (repetition time [TR], 1500 ms; echo time [TE], 28 ms; flip angle, 64°), each volume comprising 27 slices of 4 mm thickness. In-plane resolution was $3\times3$ mm$^2$ (field of view [FOV], $192\times192$ mm$^2$). Slice acquisition order was interleaved. Anatomical images were acquired using a T1-weighted magnetization-prepared rapid-acquisition gradient echo (MPRAGE) pulse sequence (TR, 2300 ms; TE, 3.08 ms; flip angle 9°; 0.89 mm$^3$ resolution; FOV, 256 mm$^2$). Participants' heads were stabilized with foam padding to minimize head movement. Auditory stimuli were presented using the Psychophysics toolbox (*Brainard, 1997*; *Pelli, 1997*). Participants were provided with MRI compatible in-ear mono earbuds (Model S14, Sensimetrics Corporation, Malden, MA), which provided the same audio input to each ear. MRI-safe passive noise-canceling headphones were placed over the earbuds for additional protection against noise.

## fMRI data preprocessing

FMRI data processing was carried out using FEAT (FMRI Expert Analysis Tool) Version 5.98, part of FSL (FMRIB's Software Library, www.fmrib.ox.ac.uk/fsl). The following procedure was applied: motion correction using MCFLIRT (*Jenkinson et al., 2002*); slice-timing correction using Fourier-space time-series phase-shifting; non-brain removal using BET (*Smith, 2002*); spatial smoothing using a Gaussian kernel of FWHM 6.0 mm; grand-mean intensity normalization of the entire 4D dataset by a single multiplicative factor; and high-pass temporal filtering (Gaussian-weighted least-squares straight line fitting, with sigma=240.0 s). The procedure for selecting the high-pass filter is described below. Preprocessed data were kept in the native functional space for all analyses, except for the within-interval searchlight analysis, which was performed across participants.

Preprocessed data were then despiked using the following procedure: for each voxel, data points that deviated from the mean by more than 5 times the inter-quartile range were removed and replaced using cubic interpolation.

## Procedure for obtaining anatomical masks: FreeSurfer and MTL segmentation

Segmentation was performed in a semi-automated fashion using the FreeSurfer image analysis suite, which is documented and available online (version 5.1; http://surfer.nmr.mgh.harvard.edu) with details described previously (e.g. *Fischl et al., 2004*; *Poppenk and Norman, 2014*). Briefly, this processing includes removal of non-brain tissue using a hybrid watershed/surface deformation procedure (*Ségonne et al., 2004*), automated Talairach transformation, intensity normalization (*Sled et al., 1998*), tessellation of the grey matter / white matter boundary, automated topology correction (*Fischl et al., 2001*; *Segonne et al., 2007*), surface deformation following intensity gradients (*Fischl and Dale, 2000*), parcellation of cortex into units based on gyral and sulcal structure (*Desikan et al., 2006*; *Fischl et al., 2004*), and creation of a variety of surface-based data, including maps of curvature and sulcal depth.

We resampled and aligned FreeSurfer segmentations of all grey matter, white matter, and cerebrospinal fluid (CSF) regions to native functional image space for use as anatomical masks. Anatomical regions were segmented according to the Desikan-Killiany Atlas (*Desikan et al., 2006*).

It is important to note that the medial temporal lobe (MTL) masks in the Desikan-Killiany Atlas do not match the canonical anatomical distinctions in the literature. For example, the parahippocampal gyrus mask comprises the medial part of the parahippocampal cortex and the posterior part of the entorhinal cortex. Therefore, instead of the FreeSurfer MTL masks, we used a probabilistic MTL atlas developed by *Hindy and Turk-Browne (2015)*. MTL regions, including perirhinal cortex, entorhinal cortex and parahippocampal cortex were defined probabilistically in MNI space, based on a database of manual MTL segmentations from a separate set of 24 participants. Manual segmentations were created on $T_2$-weighted turbo spin-echo images using anatomical landmarks (*Duvernoy, 2005*; *Carr et al., 2010*; *Schapiro et al., 2012*) and then registered to an MNI template. Finally, nonlinear registration (FNIRT; *Andersson et al., 2007*) was used to register the masks from MNI space to each participant's native space. After registration, voxels with a probability greater than 0.3 of being in a region were assigned to that ROI.

## Residualization of non-neuronal signal sources

Slow changes of respiration over time (RV) have been shown to induce robust changes in the BOLD signal (*Chang et al., 2009*) in many areas around the cerebral midline. To minimize signal change unrelated to neural activity, we used multiple linear regression to project out 3 nuisance variables from the BOLD data (*Behzadi et al., 2007*; *Silbert et al., 2014*). Nuisance regressors were:

1. the average time course of high standard deviation voxels (voxels with the top 1% largest standard deviation), as these voxels tend to have the highest fractional variance of physiological noise (e.g., cardiac and respiratory components) and are likely near blood vessels (*Behzadi et al., 2007*),
2. the average BOLD signal measured in CSF,
3. the average white matter signal.

All masks (grey matter, white matter and CSF) were obtained from the FreeSurfer segmentation procedure described above. The beneficial effects of this residualization procedure on the signal-to-noise ratio are shown in *Figure 13*. Note that this procedure was always applied after removal of low-frequency components using the high-pass filter (see below).

## Methodological challenges with analyzing pattern distance over long time scales: Selection of temporal high-pass filter cut-off

Because we were interested in the aspect of neural activity that changes slowly over time (reflecting gradual changes in context), we could not use a standard high-pass filter (with a cut-off period on the order of 120 s), as it would remove components of the signal that evolve on the scale of minutes. Thus, we were faced with the challenge of preserving slower components of the BOLD signal that reflect neural activity, while removing low-frequency components attributable to non-neuronal noise, including scanner drift and physiological noise (such as low-frequency respiratory variation and heart rate variation). Physiological noise (and a substantial component of scanner noise) was factored out

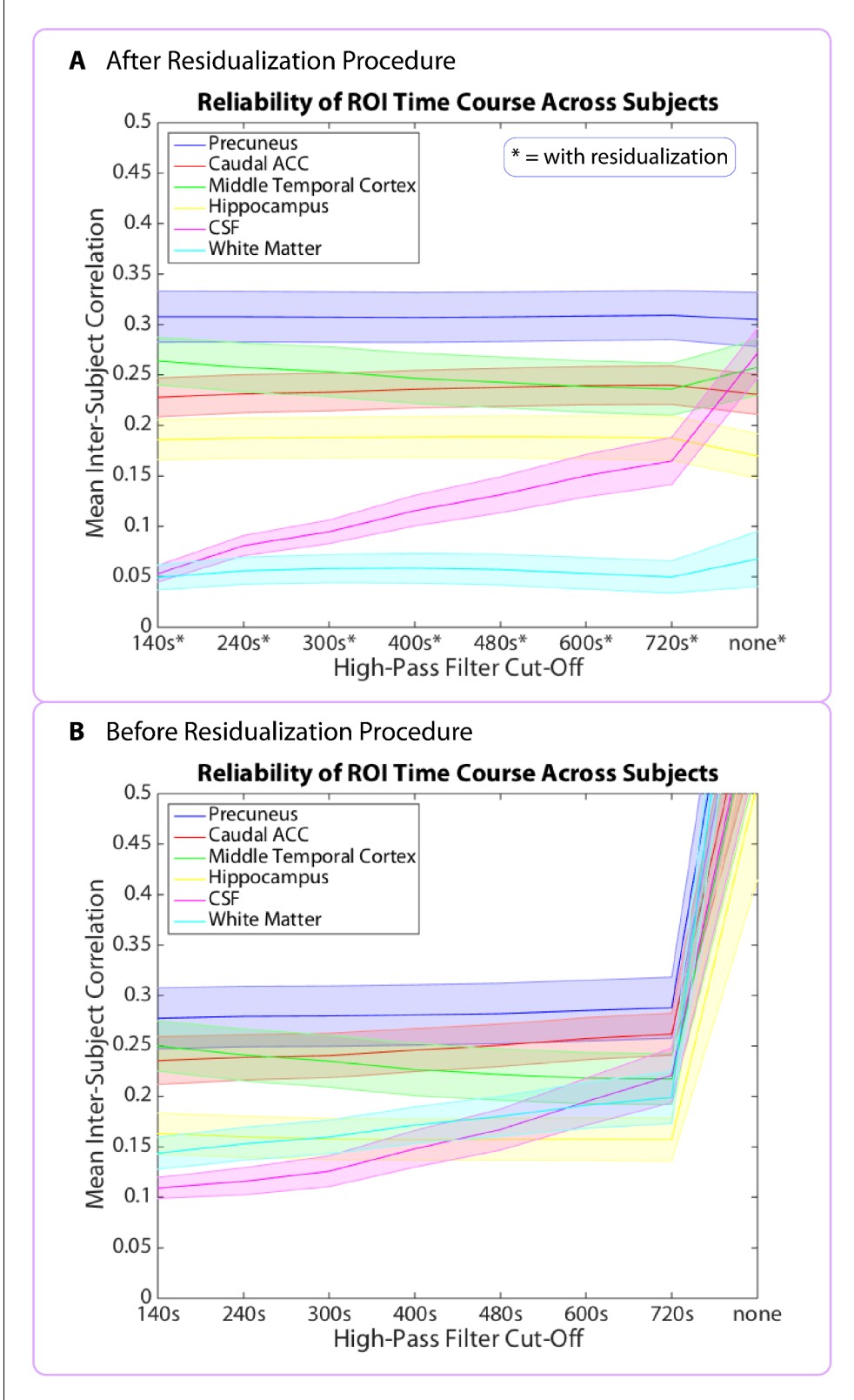

**Figure 13.** Mean inter-subject correlations (ISCs) for 6 representative brain regions as a function of the high-pass filter cut-off. Shaded error bars represent standard errors of the mean (across participants). Top panel (**A**) shows the mean ISCs after the residualization procedure has been applied (see *Residualization of non-neuronal signal sources*). The 480 s cut-off was the gentlest filter for which all of the grey matter regions listed above showed ISC values significantly above those in the CSF. Bottom panel (**B**) shows the mean ISCs prior to the residualization procedure. Without

*Figure 13 continued on next page*

Figure 13 continued

residualization, the ISCs of some grey matter regions never rise significantly above those in the white matter and CSF. Note that without high-pass filtering ('none') or residualization, all brain regions displayed spuriously high ISCs.

using the residualization procedure described above. This enabled us to select a gentler high-pass filter than is generally used in the literature.

We then performed a separate analysis to determine the optimal high-pass filter cut-off period, i.e. the lowest frequency cut-off that still enabled us to remove most of the non-neuronal noise. This analysis relies on the idea that, when participants listen to the same story or watch the same film, the signal in brain regions processing the story is highly correlated across participants (*Hasson et al., 2004*). While such correlations should not be present in CSF or white matter, spurious inter-subject correlations in these regions can arise due to low-frequency noise. In addition, listening to the same story could induce correlated motion across participants, but these correlations would also be present in CSF and white matter. Thus, we searched for a high-pass filter that could remove nonspecific correlations in CSF and white matter, while preserving correlations in brain regions known to be important for processing the stimulus. For each participant, the inter-subject correlation (ISC) of a brain region was defined as the correlation between that participant's ROI time course (averaged over voxels in that region) with the average time course of all the other participants (*Hasson et al., 2008*; *Lerner et al., 2011*).

Since the functional scan length was 1560 s (26 min), high-pass filter cut-off periods of 140 s, 240 s, 300 s, 400 s, 480 s, 600 s and 720 s were attempted. The minimal cut-off attempted, 140 s, was the cut-off used in several previous studies with naturalistic stimuli (e.g. *Lerner et al., 2011*), while 720 s represented approximately half of the scan duration and was the longest cut-off that could reasonably make a difference to data quality.

Given that roughly half the clip pairs in our time perception test were 2 min apart and the other half were 6 min apart, we hoped to find a filter that would allow us to measure pattern distances at both of these time scales. However, we were unable to find a high-pass filter that would allow us to examine activity patterns that were 6 min (360 s) apart. In order to meaningfully measure distances between neural patterns that are 360 s apart, the Nyquist theorem suggests we would need a high-pass filter cut-off of 720 s or larger. However, plotting ISC as a function of high-pass filter (*Figure 13*) showed that a cut-off like 720 s was not able to remove inter-subject correlations in the CSF, which remained of the same magnitude as those in some grey matter regions. We concluded that pattern distances at the 6-minute time scale are too confounded with low-frequency noise (as reflected in spurious correlations in the CSF), and therefore restricted our analysis to intervals that were 2 min long.

According to the Nyquist theorem, we need a filter cut-off of 4 min (240 s) or longer in order to measure distances between patterns that are 2 min apart (120 s). Out of the filters tested (240 s – 720 s), a cut-off of 480 s was selected to be the gentlest (i.e. the longest) filter that reduced the magnitude of inter-subject correlations in ventricles and CSF, such that they were significantly below the correlations in most grey matter regions.

*Figure 13* illustrates that, even for regions like the hippocampus – with relatively low inter-subject correlations – the 480 s filter cut-off, combined with the residualization procedure, succeeded at raising the grey matter ISCs significantly above those of the white matter and CSF.

## fMRI data analysis

## Within-participant correlation between pattern change and duration estimates

Our primary hypothesis was that greater pattern dissimilarity between two clips (at the time of encoding) would correlate with greater subsequent duration estimates. For each pair of clips from the time perception test, we located the TRs (volumes) corresponding to when the participant first heard those clips and extracted the activity patterns for each ROI at those time points. Since the auditory clips were between 5 s and 10 s in duration (corresponding to about 5 volumes), we

averaged the patterns over 5 consecutive TRs for every clip, with the 5-TR window centered on the middle of each clip.

We then related the pattern distance between the two clips at encoding to how much time the participant thought passed between them. Specifically, we calculated the dissimilarity (1 – Pearson correlation) between the two averaged activity patterns. The pattern dissimilarity scores for a given region were then correlated with that participant's subsequent duration estimates. This was performed separately for every ROI and searchlight (*Figure 4*). We thus obtained a Pearson correlation score for every ROI in every participant. All Pearson correlation coefficients were Fisher-transformed prior to statistical testing (*Fisher, 1915*).

To assess the reliability of the correlation across participants for a given ROI, we ran a phase-randomization procedure, which is described in detail below. The results of the phase-randomization procedure were then subjected to multiple comparisons correction.

## Removing low-confidence intervals

If a participant could not remember when in the story a particular clip had occurred, it would be difficult for them to estimate the temporal distance between that clip and another clip. It is possible that participants would invoke different retrieval strategies in such cases (for instance, they might base their duration estimates purely on the content of the clips, without recollecting their context). It is also possible that such estimates could be random guesses. To filter out guesses, we used the confidence ratings collected after the time perception test, in which participants rated how well they could remember when in the story each individual clip had occurred. Specifically, we located the participant's confidence for the two clips delimiting each temporal interval, and took the smaller of the two ratings as the confidence for that interval. We performed the main analysis relating neural drift to time estimation only on high-confidence intervals, removing pairs of clips with the lowest confidence. Since participants calibrated their confidence ratings differently (some were more prone to rate their confidence as 4/5, while others were more prone to rate it as 2/5), we picked the confidence threshold for each participant that removed at least 33% of the intervals with the lowest confidence, while preserving at least 33% of the intervals with the highest confidence. Our behavioral analysis (see *Behavioral results*) shows that participants' duration estimates were significantly more accurate for high-confidence intervals than when all intervals were included.

## Statistical analysis of correlations between pattern change and behavior

Because of the presence of long-range temporal autocorrelation in the BOLD signal (*Zarahn, 1997*), the statistical likelihood of each observed correlation (between neural distance and duration estimates) was assessed using a permutation procedure based on surrogate data. The surrogate data were generated using phase randomization (*Theiler et al., 1992*). Phase-randomized surrogates have the same autocorrelation as the original signal.

Since our analysis measures pattern change over multiple voxels, rather than the time course of a single voxel, we generated surrogate time courses of pattern change (*Figure 4—figure supplement 1* shows how that time course was obtained). Having extracted the time course of pattern change for each ROI, we applied a Fourier transform to that signal. To randomize its phases, we multiplied each complex amplitude by $e^{j\phi}$, where $\phi$ is independently chosen for each frequency from the interval $[0, 2\pi]$. In order for the inverse Fourier transform to be real (no imaginary components), we symmetrized the phases, so that $\phi(f) = -\phi(-f)$. Finally, we took the inverse Fourier transform to produce the surrogate time courses.

Each surrogate dataset was analyzed in the same manner as the empirical data: pattern dissimilarity between each pair of clips was correlated with duration estimates. Thus, we generated a distribution of 10,000 null correlations for every ROI in every participant (see *Figure 4—figure supplement 1*). As above, all correlation coefficients were Fisher-transformed to ensure that they follow a Gaussian distribution. For every ROI, we were then able to compare the empirical Pearson correlation with the distribution of null correlations. We calculated a Z-value for every participant:

$$z - value = \frac{empirical\ correlation - mean(null\ correlations)}{standard\ deviation(null\ correlations)}$$

A large positive Z-value implies that the empirical correlation is large relative to the distribution of null correlations. To assess whether the Z-values for a given ROI were reliably positive across participants, we performed a right-tailed t-test against 0. The p-values from the above t-test were then subjected to multiple comparisons correction. For anatomical ROIs (derived from the FreeSurfer and MTL atlases), we used MATLAB's fdr_bky.m function, which executes the 'two-stage' *Benjamini et al. (2006)* procedure for controlling the false discovery rate (FDR) of a family of hypothesis tests. The procedure implemented by this function is more powerful than the original *Benjamini and Hochberg (1995)* procedure when a considerable percentage of the hypotheses in the family are false. For the searchlight analysis, we controlled the family-wise error (FWE) rate, as described below.

## ROI selection

The literature reviewed above suggests that the MTL, lateral prefrontal cortex, insula, putamen and inferior parietal cortex might all process information important for inferring the duration of past events. We therefore performed an ROI analysis on the following regions, derived from both the FreeSurfer and MTL atlases: hippocampus, parahippocampal cortex, entorhinal cortex, perirhinal cortex, amygdala, superior frontal cortex, caudal and rostral middle frontal gyrus (dorsolateral prefrontal cortex), pars opercularis (frontal operculum), pars triangularis, pars orbitalis, lateral orbitofrontal cortex, frontal pole, insula, putamen and inferior parietal cortex. This resulted in an analysis on 16 regions of interest (in each hemisphere) motivated by the literature. ROIs with q-values < 0.05 (FDR) are reported as significant.

As part of an exploratory, whole-brain search, we also ran the same analysis on all grey matter regions in the Desikan-Killiany Atlas, which contained 42 regions in each hemisphere, including the ones mentioned above (see *Procedure for obtaining anatomical masks: FreeSurfer and MTL segmentation*). The complete list of regions can be found in *Figure 5—source data 1*. For the exploratory analysis, we report regions with q-values < 0.1 (FDR).

## Within-interval correlation between pattern change and duration estimates

Our main analysis verified whether the pattern distance between two clips was correlated with duration estimates in a given participant and then aggregated the results across participants. To address the concern that pattern distance between two clips might reflect only the difference in story content between those clips (rather than change in abstract factors like mental context), we performed the same analysis for a given interval across participants and aggregated the results across intervals. Since this analysis is performed within intervals, it ensures that story content is held constant across participants, such that differences in pattern distances and duration estimates are due to individual differences only. To ensure that pattern distances and duration estimates were comparable across participants, all vectors were z-scored within participants. The Pearson correlation between pattern distances and duration estimates across participants was then calculated for every 2 min interval in every ROI.

As for the within-participant analysis, this procedure was performed on high-confidence intervals. For each interval, we only included participants who had confidently recollected the temporal position of the two clips delimiting that particular interval.

The significance of each correlation score was assessed using a permutation test: 10,000 null correlations were obtained by scrambling the duration estimates across participants, such that a given participant's duration estimate was matched with a different participant's pattern distance. (Since this analysis was performed across participants, it was not necessary to generate phase-randomized pattern distance time courses – the auto-correlation in the BOLD signal for a given participant only represents a concern for the within-participant analysis.)

As above, a Z-value was obtained for every interval, reflecting the degree to which the empirical correlation was higher than the distribution of null correlations. Finally, a right-tailed t-test was performed to assess whether a given ROI's Z-values were reliably positive across intervals. The p-values from this t-test were subjected to multiple comparisons correction using FDR.

To compare effect sizes between the within-interval and within-participants analyses, we calculated Cohen's *d* for a region as:

$$Cohen's\,d = \frac{mean\,r\,(across\,participants\,or\,intervals)}{standard\,deviation\,r}$$

where $r$ is the Pearson's correlation between pattern distance and duration estimates. (Using the Z-values derived from the permutation procedures rather than the raw correlation coefficients yielded practically identical results.)

## Mixed-effects model accounting for naïve duration estimates

We analyzed our data using a hierarchical linear regression model (*Gelman and Hill, 2006*). Known in different fields as hierarchical, mixed, or multi-level models, such regressions correctly account for non-independence of repeated observations of the same subject and stimulus (in our case, interval). In doing this, they estimate the population effects (coefficients) of interest, even assuming that individual subjects or items (henceforth, collectively 'groups') may have idiosyncratic perturbations from the population and that those perturbations may be correlated within a group. They are a generalization of approaches that treat all observations as independent (e.g. t-test, ANOVA, linear regression), as well as of approaches that can take into account the non-independence across a single grouping factor (e.g. repeated-measures ANOVA), and are more conservative than any of the above (*Barr et al., 2013*). (More precisely, methods that do assume observation independence are anti-conservative in the presence of correlated observations.)

Formally, the model is the following:

$$y_i = X_i(\beta + s_{j[i]} + m_{k[i]}) + \epsilon$$
$$s_j \sim N(0, \Sigma_s), \quad m_k \sim N(0, \Sigma_M), \quad \epsilon \sim N(0, \sigma)$$

Here, $y_i$ is the $i$th observed duration judgment, $X_i$ is a matrix of predictors (neural pattern distance) and covariates (naïve duration estimates), $\beta_i$ is a vector of coefficients (as in conventional linear regression), $j[i]$ is the subject of the $i$th observation, so that $s_{j[i]}$ is a subject-specific perturbation of all of the coefficients, and $m_{k[i]}$ is similarly an item-specific perturbation of the coefficients.

This model is undefined when either the subject or item effects approach zero (either because there is truly no variability, or more realistically when there is insufficient data to estimate this variability). Since such rich models often fail to converge or approach singularity given typical psychological datasets (*Bates et al., 2015a*), we imposed a weak Wishart prior on the group covariances, which regularizes the model away from singularity (*Chung et al., 2015*). This weak, boundary-avoiding prior on our random effects covariance structure regularizes the model towards simpler random effects structures unless the data suggests otherwise (*Chung et al., 2015*). All models converged under this prior. This fitting procedure was implemented using the R package blme (*Chung et al., 2013*), which extends the lme4 package (*Bates et al., 2015b*) and performs maximum-a-posteriori estimation of linear mixed-effects models.

Please note that we also verified that our results were replicable using an alternative fitting procedure suggested by *Bates et al. (2015a)*. We used the lme4 package to fit the 'maximal' model (in the sense of *Barr et al., 2013*) and removed zero-variance random effects terms until the model converged and until the estimated random effects covariance matrix was full-rank, indicating a non-degenerate estimate. We obtained highly consistent results using both fitting procedures. In the *Results* section, we report only the first procedure, which has been found to be more conservative (*Chung et al., 2015*). *Chung et al. (2015)* report: 'Uncertainty for the fixed coefficients is less underestimated than under classical ML or restricted maximum likelihood estimation.' Indeed, our effects were very slightly stronger using the second procedure (*Bates et al., 2015a*). Both sets of results can be found in *Figure 7—source data 1*.

Finally, the duration estimates are bounded at zero and positively skewed, which resulted in heteroskedastic residuals. To mitigate this, we power-transformed the duration estimates using the Box-Cox power transformation (*Box and Cox, 1964*). We picked the exponent $\lambda$ for each model by maximizing the profile likelihood in a model without group effects (though see e.g. *Gurka et al. (2006)* for an extension to the hierarchical case).

In R formula notation, a model of the following form was fit to the data from each region of interest:

$$Transformed\,Duration\,Estimates \sim 1 + Naive\,Estimates + Neural\,Pattern\,Distance$$
$$+ (1 + Naive\,Estimates + Neural\,Pattern\,Distance \mid Subject)$$
$$+ (1 + Neural\,Pattern\,Distance \mid Interval)$$

Please note that participants from the original experiment could not be 'matched' with participants from the naïve experiment. For this reason, naïve duration estimates were group-averaged and the mean vector of naïve estimates was placed as a covariate in the model. The above formula shows that the slope of the relationship between naïve estimates and original duration estimates was allowed to vary by subject (i.e. each participant's duration estimates might be differently related to the naïve group mean). On the other hand, the slope for naïve estimates could not vary by interval, since naïve estimates did not vary by subject.

We computed 0.95 confidence intervals of $\beta$ using the asymptotic Gaussian approximation (called the 'Wald approximation' in lme4) based on the estimated local curvature of the likelihood surface. Since this approximation is anti-conservative (it assumes infinite data and no model misspecification), we then computed a more conservative parametric bootstrap interval for the intervals that did not include zero. Effects whose interval does not overlap with 0 are significant at the conventional $\alpha$=0.05 level.

Note that all of the above choices (including the choice of fitting procedure and the power transform of the data) are conservative relative to their alternatives. For instance, prior to power-transforming the duration estimates, the fixed effects of neural pattern distance were estimated to be stronger (as reported in *Figure 7—source data 1*.) These alternative analyses revealed additional significant regions that are either false positives or regions we lack the power to detect.

## Whole-brain searchlights

In addition to using anatomical ROIs, we ran a cubic searchlight throughout the entire brain. The same analysis as described above was performed for every searchlight, and the Z-value for each searchlight was assigned to the center voxel.

The within-participant analysis was performed in native functional space, and each cubic searchlight contained 3x3x3 (27) voxels. To aggregate the results across participants, each participant's Z-value map was transformed to standard MNI space and down-sampled to 3 mm to reflect the resolution of the original data.

The within-interval analysis was performed in 3 mm MNI space, in order to match the searchlights across participants. Since this transformation approximately doubles the number of brain voxels, we ran cubic searchlights of radius 2 with 5x5x5 (125) voxels through the entire brain. Neural pattern distance was not calculated for searchlights on the very edge of the brain with fewer than 25 voxels, in order to reduce noise from overly small patterns. We also excluded a searchlight location if fewer than 5 participants had brain voxels in that location.

Family-wise error rate was controlled using FSL's *randomise* function (version 5.0.4, *Winkler et al., 2014*). An uncorrected p-value image was first generated, reflecting voxel-wise (searchlight) reliability across participants or intervals. The significance of supra-threshold clusters (defined by the cluster-forming threshold, p<0.01) was then assessed by cluster mass. Specifically, a corrected p-value was assigned to each cluster by assessing its cluster mass with respect to the null distribution of the maximum cluster mass during 10,000 permutation simulations (*Hayasaka and Nichols, 2003*; *Nichols and Holmes, 2002*). Cluster coordinates are reported in MNI space, and cluster size reflects the number of voxels in 3x3x3mm MNI space.

## Comparing speed of pattern change across brain regions

If the brain regions that showed significant effects in our main analysis represent mental context, then the pattern of activity in these regions should change more slowly over time than the patterns in regions representing sensory information. To quantify the speed of pattern change in a given ROI, we obtained the correlation of the pattern at every time point (TR) with itself at every other time point. (As for our main analysis, the BOLD time course of every voxel was smoothed using a moving average filter of 5 TRs. This temporal smoothing was used as a de-noising technique and did not affect the results.) We then averaged the auto-correlation curves across TRs to obtain a mean auto-correlation function for every region in every participant. The more rapidly a pattern changes over

time, the more sharply the auto-correlation should decrease as we move away from 0. To quantify this, we defined the Full-Width Half-Maximum (FWHM) of the auto-correlation curve as the number of time points (TRs) for which the auto-correlation was equal to or greater than half its maximum value (the maximum was always 1.)

To compare the speed of pattern change in the regions we found (right entorhinal cortex and left caudal ACC) with regions involved in auditory and language processing, we performed a paired Wilcoxon signed rank test on the FWHM values across participants. The p-values from this test were subjected to multiple comparisons correction using FDR.

Since the anatomical masks we used varied substantially in size, we sought to ensure that differences in the speed of pattern change were not due to differences in ROI size. For this purpose, we performed the same analysis after regressing the vector of ROI sizes out of the vector of FWHM values for every participant.

Since the above regression would only account for a linear effect of ROI size on the speed of pattern change, we additionally performed a univariate analysis that calculated the auto-correlation function for each voxel individually. The auto-correlation curve was obtained by correlating the BOLD time course of every voxel with itself at all possible lags. The mean auto-correlation for an ROI was obtained by averaging the auto-correlation curves across all the voxels in that ROI. The FWHM values were then calculated in the same manner as above for every ROI in every participant.

### Replication of *Jenkins and Ranganath (2010)* 'coarse temporal memory' fMRI analysis

As in *Jenkins and Ranganath (2010)*, we correlated each voxel's activity during encoding of a clip with the accuracy of a participant's placement of that clip on the timeline. Voxel activity was averaged over a 5-TR window centered on the mid-point of the clip. For each participant, the estimated clip position on the timeline was regressed against actual position. Accuracy was defined as the negative error, which was the absolute value of the residual for a clip. Within participants, voxel activity was then correlated with accuracy across all clips, and the Pearson's *r* score was Fisher-transformed. As for the within-participant searchlight analysis, transformed *r* score maps were registered to 3mm MNI space, and FSL's *randomise* was used to control the FWE rate.

## Acknowledgements

We would like to thank Lucy Lin for her assistance with data collection for the event boundary experiment. We would like to thank Erez Simony, Lili Sahakyan, Mariam Aly, Anna Schapiro and Michael Chow for their advice on data analysis and preprocessing, as well as helpful discussion.

## Additional information

### Funding

| Funder | Grant reference number | Author |
|---|---|---|
| National Institutes of Health | Early Stage Investigator, R01-MH094480 | Uri Hasson |
| John Templeton Foundation | Proposal 36751 | Olga Lositsky Kenneth A Norman |
| National Institutes of Health | Training Grant, 2T32MH065214 | Olga Lositsky Janice Chen |

The funders had no role in study design, data collection and interpretation, or the decision to submit the work for publication.

### Author contributions

OL, Conception and design, Acquisition of data, Analysis and interpretation of data, Drafting or revising the article; JC, DT, CJH, UH, KAN, Conception and design, Analysis and interpretation of data, Drafting or revising the article; MS, JLP, Analysis and interpretation of data, Drafting or revising the article

## Author ORCIDs

Olga Lositsky, http://orcid.org/0000-0001-7089-4474
Janice Chen, http://orcid.org/0000-0003-4511-4725
Christopher J Honey, http://orcid.org/0000-0002-0745-5089
Jordan L Poppenk, http://orcid.org/0000-0002-3315-5098
Uri Hasson, http://orcid.org/0000-0002-3599-7168
Kenneth A Norman, http://orcid.org/0000-0002-5887-9682

## Ethics

Human subjects: All parts of the experimental procedure were approved by the Princeton Institutional Review Board under Protocol #5516. All participants were screened to ensure no neurological or psychiatric disorders. Written informed consent, and consent to publish, was obtained for all participants in accordance with the Princeton Institutional Review Board regulations.

## Additional files

### Major datasets

The following dataset was generated:

| Author(s) | Year | Dataset title | Dataset URL | Database, license, and accessibility information |
|---|---|---|---|---|
| Lositsky O, Chen J, Toker D, Honey CJ, Hasson U, Norman KA | 2016 | Neural pattern change during encoding of a narrative predicts retrospective duration estimates | http://dataspace.princeton.edu/jspui/handle/88435/dsp011n79h6771 | Publicly available at the Princeton dataspace |

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
