## [Decision Letter]

[Editors’ note: a previous version of this study was rejected after peer review, but the authors submitted for reconsideration. The first decision letter after peer review is shown below.]

Thank you for choosing to send your work entitled "Neural pattern change during encoding of a narrative predicts retrospective duration estimates" for consideration at *eLife*. Your full submission has been evaluated by Timothy Behrens (Senior editor) and two peer reviewers and a member of our Board of Reviewing Editors, and the decision was reached after discussions between the reviewers. Based on our discussions and the individual reviews below, we regret to inform you that your work will not be considered further for publication in *eLife*.

Reviewer #1:

The logic of the paper is that in some regions (notably EC and pars orbitalis) the RSA distance between the multivoxel response at two moments predicts the time subjects judge between those two events later on. The authors argue that this result suggests that retrospective time judgements depend on a gradually changing state of temporal context that resides in these regions. If we take its conclusions at face value, this paper would make an important contribution. The paper is potentially an important advance over previous studies because it uses realistic stimuli.

Unfortunately, I don't quite accept the conclusions. The fundamental problem is that I can imagine obtaining the result without any memory demands whatsoever. Imagine that the participants were played audio clips from a radio drama and asked to judge how far apart they were in the show. Let's say for one pair of clips both have the sound of the ocean in the background and the same speakers speaking. The other pair of clips does not sound alike---different people are speaking in different locations. Which of these pairs would be judged to be closer in time? Now, insofar as an aspect of brain activity measures any property of the clips (power spectrum, semantic content, etc), we would readily expect that brain activity to correctly categorize the pairs of clips. But this is certainly not a memory effect as by construction, there is no actual memory for anything. This account seems to naturally account for the finding that many many brain regions show a tendency towards a correlation (Figure 5), although most of them do not reach significance.

I would feel much better about accepting the conclusions if, rather than assessing significance relative to chance, the analysis was done relative to some control region that ought to be sensitive to auditory and/or semantic similarity. I am not enough of an expert in the auditory system (or fMRI more broadly) to suggest a specific comparison region, but as it is I think the conclusions either need to be significantly moderated or the empirical support for those conclusions needs to be stronger.

Reviewer #2:

Using multivoxel pattern similarity, the authors find that right entorhinal cortex, right ATL, right pars orbitalis and left ACC show patterns of activity that correlate with cued retrospective duration judgments while keeping objective duration constant. They show this using both an ROI and searchlight approach and find an overlapping, though not completely identical, set of regions which they attribute to differences between the two methods in size, shape, and respect of anatomical boundaries. The experiment is interesting and methodologically sound but would be more impactful if the authors did more work to understand what is driving their effect. As is, the authors don't do much to make the reader excited about the findings or to better differentiate it from prior related work.

For example, Figure 2—figure supplement 1 suggests that certain pairs of story clips are consistently rated as closer together versus further apart. There appears to be no attempt to characterize what features of the story drive this effect. Moreover, it is unclear how much such features may produce the effect of neural dissimilarity correlating with greater distance judgments. For instance, if two clips with two different sets of characters are rated as further apart that two clips with the same set of characters, the regions that show dissimilarity scaling with subjective duration may be those sensitive specifically to characters rather than context more broadly. Thus an alternate explanation for their results is that the regions that show their effect are just sensitive to the content of the story that can produce both neural dissimilarity and greater duration ratings. The authors should discuss/examine this alternative.

One analysis that might support their account of effects being due to gradual change in context tracking regions would be to split their 2 min interval into 20-30s chunks and see if the change is indeed gradual. Otherwise the dissimilarity measure could be simply a result in differences in evoked activity between the two time points, which would be more likely if the effects were due to content sensitivity.

It would be informative to know whether the pattern similarity values in their regions correlate with each other as they should if they are tracking the same context state representation?

Did duration ratings change as a function of position in the story? One could imagine that overall recency would have an effect on duration judgments. And if so, did pattern similarity values change?

[Editors’ note: what now follows is the decision letter after the authors submitted for further consideration.]

Thank you for submitting your article "Neural pattern change during encoding of a narrative predicts retrospective duration estimates" for consideration by *eLife*. Your article has been reviewed by two peer reviewers, and the evaluation has been overseen by a Reviewing Editor and Timothy Behrens as the Senior Editor. The following individuals involved in review of your submission have agreed to reveal their identity: Marc Howard (Reviewer #1); Lila Davachi (Reviewer #2).

The reviewers have discussed the reviews with one another and the Reviewing Editor has drafted this decision to help you prepare a revised submission.

1) Please analyze and report univariate data – to see if they replicate Jenkins and Ranganath – or not.

2) Focus the paper more on the host of regions that show a slowly changing neural signal over time – instead of focusing on entorhinal and pars orbitalis (area 47).

3) Please discuss the divergence in their neural data from the primacy and recency effects in behavior.

4) Please test control duration judgments vs experimental duration judgments directly. If there's a reliable effect, then at least part of what they're calling mental context is most likely not mental context. This outcome probably does not lead to an *eLife* paper.

5) Given that there isn't a reliable effect, the reviewers need to be convinced that this lack of an effect is meaningful. One way to do this would be to do a power analysis. Another way would be to place a confidence interval on the correlation and show that the correlation, while possibly non-zero, would have to be so small we wouldn't care about it. Even better if you can argue it would have to be so small it couldn't account for the correlation between the experimental judgments and the drift. There are also fancy ways to approach this (e.g., Bayesian inference). In sum, the authors need to make a positive case for the null if they want to argue that this is a memory effect rather than some property of their stimuli.

Reviewer #1

On the previous round of review, my major concern was that the change attributed to putative contextual drift that correlates with duration judgments could more simply be attributed to perceptual/semantic differences in the patterns themselves. Real world stimuli that unfold in time are autocorrelated over just about every time scale. Compounding the problem, it is impossible to measure the similarity on all relevant dimensions. This revision makes a substantive attempt to argue against the perceptual hypothesis, there are two behavioral controls that attempt to address the question of whether the results attributed to contextual change could be driven by perceptual effects. To summarize my reaction to the revision, while the controls make for a stronger case than the previous submission, I am not convinced that these controls address the concern in a satisfactory way. I suggest additional analyses with the existing data that could clarify this point. If this concern were resolved (which is not at all clear) the manuscript would result in a very nice contribution.

The first control asks subjects to describe event boundaries during presentation of the story. This allows a rough estimate of the change in context between the two clips. Indeed the number of event boundaries predicted duration judgments by the original participants. The assumption seems to be that there is no perceptual/semantic similarity across event boundaries, but is not an assumption that I can accept. Imagine a story where Alice and Betty have a discussion at the beach. Then there's an event boundary and Alice and Betty move to the coffee shop. Then there's another event boundary as Alice leaves and Betty and Chris have a conversation in the coffee shop. The number of event boundaries correlates with the overlap of perceptual/semantic features present in the scene. More broadly, if the perceptual/semantic content is autocorrelated over long time scales (and it almost surely is) and if event boundaries are a proxy for abrupt drops in the autocorrelation, then number of event boundaries really ought to predict perceptual/semantic similarity of the available features. So this control is not at all convincing.

In the second control, a group of naive subjects are asked to rate the similarity of the clips. There is no evidence that their ratings correspond to the number of event boundaries between the clips. The suggestion is that because number of event boundaries indexes contextual change (but not presumably perceptual/semantic similarity), the null result requires us to accept that there is no difference in the similarity of the two clips. Leaving aside for a moment the issue of asking the reader to accept the null (which is a really serious problem!), this is kind of an indirect test of what we're really after. The finding is that duration judgments of the fMRI subjects correlate with number of event boundaries whereas the judgments of the naive controls do not correlate with number of event boundaries. Why not just ask whether the judgments of the fMRI subjects correlate with the judgments of the naive subjects? If they do, then there is no way to argue that the change in the multivoxel signal is due to contextual drift per se. It might be possible to partial out the effect attributable to the naive subjects' judgments. If there is not a correlation, then the authors still have to successfully argue for the null, but it's at least a clean and direct (and much more sensitive!) test of the question of interest.

Reviewer #2

This revision has been responsive to prior concerns about whether visual or semantic dissimilarity during temporal memory judgments could be used to infer how far apart the clips had been presented during encoding. The authors conducted behavioral analyses to show that distance judgments were related to listening to the story and could not be deduced based on the test stimuli alone. They show that the number of event boundaries experienced in between the test stimuli also modulated distance judgments. These new results remove any doubts that the reported effects are not driven by visual confounds.

However, I am somehow not that excited about the new ms as it now provides a list of more regions that show pattern change related to distance judgments – much of the medial temporal lobe, frontal cortex, anterior temporal cortex, ACC… given the effects are more widespread, the laser focus on entorhinal and pars orbitalis makes the paper not easy to digest. Is this a general broad signal? Or is it focused?

Also new data that has been added in response to other concerns that now raise some skepticism. The most intrusive is the fact that distance judgments vary predictably by 'list' position – events early in the audiovisual recording are remembered as farther apart than those later in the tape. (see Figure 11 – top panel). However, pattern similarity estimates do not track this behavioral effect. This result raises questions about why, if entorhinal and frontal cortex are representing temporal context signal, would they not also some what mirror the behavioral judgments. I could imagine that context representations may play less of a role as item memory fades? This is not discussed but should be for the authors' views on this to be clear. Otherwise, the impact of the final result is unclear.

In my original review, I had requested that they examine whether univariate activity was related to temporal memory success. The did run that analysis but only in entorhinal cortex and pars orbitalis -and do not see any effects but beg off reporting it since they did not have *a priori* predictions about it. however, published work (Jenkins and Ranganath) has shown that univariate activity in more dorsal parts of lateral frontal cortex is related to coarse temporal memory judgments. so there is a clear precedent for this effect. I am not sure why they say they did not have that prediction but this analysis, even if the results DO NOT show a univariate effect would be informative and could even bolster their conclusions that patterns across time, rather than activity to any single event, is a better predictor of temporal memory judgments.

---

## [Author Response]

[Editors’ note: the author responses to the first round of peer review follow.]

*Reviewer #1:*

*The logic of the paper is that in some regions (notably EC and pars orbitalis) the RSA distance between the multivoxel response at two moments predicts the time subjects judge between those two events later on. The authors argue that this result suggests that retrospective time judgements depend on a gradually changing state of temporal context that resides in these regions. If we take its conclusions at face value, this paper would make an important contribution. The paper is potentially an important advance over previous studies because it uses realistic stimuli.*

[…]

*I would feel much better about accepting the conclusions if, rather than assessing significance relative to chance, the analysis was done relative to some control region that ought to be sensitive to auditory and/or semantic similarity. I am not enough of an expert in the auditory system (or fMRI more broadly) to suggest a specific comparison region, but as it is I think the conclusions either need to be significantly moderated or the empirical support for those conclusions needs to be stronger.*

We thank the reviewer for raising these essential questions. In order to address them, we conducted two new behavioral experiments, as well as two new analyses of the neural data:

1) We show that our participants' duration estimates correlate strongly with the number of event boundaries between two clips, suggesting that their estimates were influenced by their memory for the content of the story in between two clips (rather than the similarity between the two clips alone).

2) A separate group of participants was asked to complete the same time perception test without first listening to the story. Since these "naive participants" had no memory of the story, they could only base their duration estimates on the similarity between the two clips. We show that duration estimates from these naive participants do not correlate with the number of event boundaries between two clips, proving that the intervening content between clips does not influence duration estimates when participants have no memory of the story.

3) A related concern was that neural pattern change might be driven by the perceptual and semantic dissimilarity of the clips, rather than the degree of contextual drift between the clips. To address this, we performed a within-interval version of our main ROI analysis. This analysis holds constant the two clips whose pattern distance is being measured. We show that individual differences in neural pattern change for a given pair of clips correlate with individual differences in duration estimates in the right entorhinal cortex and right pars orbitalis, as well as other regions that had been sub-threshold in our main analysis. Thus, pattern change in these regions correlates with duration estimates even when the perceptual and semantic content of the two clips is held constant. If neural pattern change were being driven by story content, we would have expected the effect to be larger for the across-interval, within-participants analysis (where story content differed across intervals) than for the across-participants, within- interval version of the analysis (where story content is held constant). The fact that the effect was similar in size for the two analyses suggests that story content is not a major factor driving the observed correlation between neural pattern change and duration estimates.

4) We show that patterns of activity in entorhinal cortex and pars orbitalis change significantly more slowly over time than patterns in cortical regions implicated in auditory and language processing, suggesting that they may integrate information over longer time scales.

We believe, and we hope the reviewer will agree, that these analyses directly address and alleviate the concern that our results could be obtained "without any memory demands whatsoever".

A summary of the new analyses is now presented in a section titled "Factors Driving the Correlation between Pattern Change and Duration Estimates":

“[…]we conducted two control behavioral studies. One group of participants indicated when event boundaries were occurring in the story. […]Moreover, pattern change in the right entorhinal cortex correlates highly with pattern change in the right pars orbitalis, suggesting that the two regions may cooperate to represent different facets of a unified, slowly changing context signal.”

A more detailed description of each analysis follows this section.

*Reviewer #2:*

*Using multivoxel pattern similarity, the authors find that right entorhinal cortex, right ATL, right pars orbitalis and left ACC show patterns of activity that correlate with cued retrospective duration judgments while keeping objective duration constant. They show this using both an ROI and searchlight approach and find an overlapping, though not completely identical, set of regions which they attribute to differences between the two methods in size, shape, and respect of anatomical boundaries. The experiment is interesting and methodologically sound but would be more impactful if the authors did more work to understand what is driving their effect. As is, the authors don't do much to make the reader excited about the findings or to better differentiate it from prior related work.*

We are grateful to the reviewer for their comments on how to increase the impact of the work. As described below, we have made several substantial changes to the paper to further specify what is driving our effect.

For example, Figure 2—figure supplement 1 suggests that certain pairs of story clips are consistently rated as closer together versus further apart. There appears to be no attempt to characterize what features of the story drive this effect. Moreover, it is unclear how much such features may produce the effect of neural dissimilarity correlating with greater distance judgments. For instance, if two clips with two different sets of characters are rated as further apart that two clips with the same set of characters, the regions that show dissimilarity scaling with subjective duration may be those sensitive specifically to characters rather than context more broadly. Thus an alternate explanation for their results is that the regions that show their effect are just sensitive to the content of the story that can produce both neural dissimilarity and greater duration ratings. The authors should discuss/examine this alternative.

We thank the reviewer for pointing out this important concern. Reviewer #1 raised essentially the same question: are the regions that show dissimilarity scaling with subjective duration sensitive to story content (e.g., which characters are present), rather than to context more broadly? Please see our response to Reviewer # 1 above (Major comment #1), in which we describe new behavioral control experiments and new neural analyses. The behavioral controls suggest that the number of event boundaries between two clips is a strong driver of duration estimates (but only for participants who have heard the story), and may be the reason why duration estimates are so consistent across participants. We also present a new neural analysis showing that all of the regions we found display a significant correlation between neural pattern dissimilarity and duration estimates across participants for a given pair of clips, in other words, even when the (objective) content of the story is held constant.

A summary of the new analyses is now presented in the section titled "Factors Driving the Correlation between Pattern Change and Duration Estimates"; a more detailed description of each analysis follows this section.

*One analysis that might support their account of effects being due to gradual change in context tracking regions would be to split their 2 min interval into 20-30s chunks and see if the change is indeed gradual. Otherwise the dissimilarity measure could be simply a result in differences in evoked activity between the two time points, which would be more likely if the effects were due to content sensitivity.*

We greatly appreciated the reviewer's suggestion to "split the 2 min interval into 20-30s chunks and see if the change is indeed gradual", and decided to expand on this idea to analyze the speed of pattern change across the entire story timecourse.

We quantified the speed of pattern change by calculating the auto-correlation of the pattern in a given region for every time point (TR) and averaging across time points to obtain a mean auto- correlation curve. The full-width half-maximum (FWHM) of this curve was taken as a measure of pattern change speed (the wider the peak of this curve is, the more gradually the pattern changes over time). We found that patterns of activity in the right entorhinal cortex and right pars orbitalis changed significantly more slowly than patterns in the right transverse temporal cortex (primary auditory cortex), right banks of the superior temporal sulcus and right superior temporal cortex (involved in auditory and language processing). Importantly, we also found that the right entorhinal cortex and right pars orbitalis, along with neighboring regions in the temporal pole, medial temporal lobe, orbitofrontal cortex and frontal pole, had the highest FWHMs (slowest pattern change) in the entire brain.

These results are now presented in the manuscript (section titled "Patterns of activity in entorhinal cortex and pars orbitalis change slowly over time").

Since the anatomical masks used in the above analysis were of different sizes, we performed two control analyses to ensure that differences in the speed of pattern change were not due to differences in ROI size.

First, we regressed the vector of ROI size out of the vector of FWHM values across regions for every participant. This modified analysis replicated the results reported above: the entorhinal cortex, pars orbitalis, as well as other ROIs in the anterior temporal lobe, medial temporal lobe and orbitofrontal cortex, still had the slowest pattern change in the brain, and significantly slower than in primary auditory cortex. These results are reported in the manuscript.

Second, we performed a univariate version of the above analysis by calculating the auto- correlation function of each voxel individually, averaging the auto-correlation curves across all voxels of a given ROI and then computing the FWHM value for the average curve. The univariate analysis replicated the above findings, and showed that the right entorhinal cortex ROI had the slowest changing voxels of all the regions in our atlas. These results are reported in the manuscript.

Taken together, we feel that these new analyses provide strong support for our interpretation that entorhinal cortex and pars orbitalis process information that changes gradually over time.

*It would be informative to know whether the pattern similarity values in their regions correlate with each other as they should if they are tracking the same context state representation?*

We agree with the reviewer that regions tracking the same context state representation should have correlated pattern change values across two-minute intervals. To explore this, we extracted the pattern dissimilarity for each of the 24 pairs of clips (which were 2 min apart) and averaged the vectors across participants. The correlation between the mean pattern distance vectors in the right entorhinal cortex and right pars orbitalis was r = 0.73. In order to interpret the magnitude of this correlation, we also calculated the correlation between every possible pair of mean pattern distance vectors (for all 84 anatomical masks). This resulted in a distribution of 3486 correlations for every possible pair of regions ((84 x 84 – 84) I 2 = 3486).

Out of 3486 pairs of regions, only 242 exhibited a correlation that was higher than the one observed between the right entorhinal and right pars orbitalis. Thus, the correlation between the pattern distances in these two regions is higher than for 93% of region pairs.

A phase randomization procedure showed that the likelihood of obtaining a correlation of this magnitude by chance – given the auto-correlation in the pattern change vectors – was p=0.0011.

The strong correlation in pattern change between the two regions suggests that they may cooperate to represent different facets of a unified, slowly changing context signal.

This analysis is now reported in greater detail in the manuscript.

*Did duration ratings change as a function of position in the story? One could imagine that overall recency would have an effect on duration judgments. And if so, did pattern similarity values change?*

Duration estimates did change as a function of position in the story, with earlier intervals being estimated as longer than later intervals (Figure 11). The correlation between the estimated duration of an interval and its time in the story was consistently negative across participants (M= -0.40, SD= 0.22; t(16)= -7.59, p<0.00001). These results replicate the positive time-order effect, which is the finding that people judge earlier durations in a series of durations to be longer than later durations (Block, 1982, 1985; Brown & Stubbs, 1988). The effect has been interpreted to mean that context changes more rapidly at the start of a novel episode (Block, 1982, 1986).

Interestingly, the pattern dissimilarity values in right entorhinal cortex and right pars orbitalis did not exhibit the same overall decrease across time. In fact, there was no consistent correlation between pattern change during an interval and its time in the story for the right entorhinal cortex (M=0.03, SD=0.21; t(16)= 0.65; p=0.53) or the right pars orbitalis (M=-0.10, SD=0.22; t(16)= -1.83, p=0.09). These results suggest that the relationship between duration estimates and pattern dissimilarity in these regions was not driven by a shared linear trend. Rather, it seems that pattern dissimilarity in these regions correlated with more fine-grained variations in the estimated durations of nearby intervals (Figure 11).

This analysis is now presented in the manuscript.

[Editors' note: the author responses to the re-review follow.]

*The reviewers have discussed the reviews with one another and the Reviewing Editor has drafted this decision to help you prepare a revised submission.*

*1) Please analyze and report univariate data – to see if they replicate Jenkins and Ranganath – or not.*

*2) Focus the paper more on the host of regions that show a slowly changing neural signal over time – instead of focusing on entorhinal and pars orbitalis (area 47).*

*3) Please discuss the divergence in their neural data from the primacy and recency effects in behavior.*

*4) Please test control duration judgments vs experimental duration judgments directly. If there's a reliable effect, then at least part of what they're calling mental context is most likely not mental context. This outcome probably does not lead to an eLife paper.*

*5) Given that there isn't a reliable effect, the reviewers need to be convinced that this lack of an effect is meaningful. One way to do this would be to do a power analysis. Another way would be to place a confidence interval on the correlation and show that the correlation, while possibly non-zero, would have to be so small we wouldn't care about it. Even better if you can argue it would have to be so small it couldn't account for the correlation between the experimental judgments and the drift. There are also fancy ways to approach this (e.g., Bayesian inference). In sum, the authors need to make a positive case for the null if they want to argue that this is a memory effect rather than some property of their stimuli.*

We would like to thank the reviewers for their tremendously helpful comments, which have guided our revisions and inspired us to add new analyses that we feel have substantially improved the rigor of our contribution. These revisions have also enabled us to reorganize our manuscript into a far more coherent structure that helps highlight the consistency of our findings across analyses.

The following is a summary of the most important changes.

Reviewer 1 was concerned that, if our original behavioral data is correlated with the behavioral data from the control (naïve) group who had not heard the story, then a component of our original behavior could be correlated with the perceptual or semantic similarity between clip pairs and that this component could be driving the correlation with neural pattern change. We address this concern in the following ways:

First, we emphasize the importance of the within-interval analysis, which correlates individual differences in subjective duration for a pair of clips with individual differences in neural pattern drift. This analysis holds constant the objective similarity of the two clips and leverages variance across participants. In addition to the ROI analysis from the previous version of the manuscript, we report a new searchlight version of the within-interval analysis, demonstrating that a cluster in the right anterior temporal lobe, overlapping with the right entorhinal region found in the ROI analysis, is significant even when the objective similarity between two clips is controlled for.

Second, we perform a highly conservative mixed-effects version of the ROI analysis. For each ROI, we fit a model that estimates the population-level effect of neural pattern distance on duration estimates, while controlling for individual variability between participants and between clip pairs. This analysis combines the virtues of both the within-participant and within-interval analyses. We show that the right entorhinal cortex and left caudal ACC exhibit confidence intervals that do not include 0, even when the most conservative fitting procedure and power transform of the behavioral data are applied. Moreover, we show that these effects are not weakened by including the mean duration estimates from the control (naïve) group in the model.

The new analyses above also help us to address the concern from Reviewer 2 that, while we reported effects in a distributed set of brain regions, our discussion focused on only two or three of those regions. The revised manuscript explicitly synthesizes the findings from all of the above analyses in a section entitled “Comparing Results from ROI and Searchlight Analyses”. The synthesis shows that regions of the right anterior temporal lobe, peaking in the right entorhinal cortex, emerge consistently across the within-participant and within-interval versions of the ROI and searchlight analyses, as well as the mixed-effects ROI analysis.

As requested by Reviewer 2, in the new manuscript, we discuss in detail the lack of linear decrease in neural pattern change with time in the story. Raw fMRI data prior to high-pass filtering shows that if such a trend were present, it would be obscured by an existing linear trend in the opposite direction, which seems to be caused by non-neuronal scanner artifacts.

Finally, we report an exciting new replication of the Jenkins and Ranganath (2010) coarse temporal memory analysis. After performing a whole-brain univariate analysis, we find a significant correlation between activity during encoding of a clip and the participant’s later accuracy in placing that clip on the timeline of the story in extensive clusters of the lateral prefrontal cortex (DL-PFC) and dorsomedial PFC, as well as sub-threshold clusters in the medial parietal. Our results suggest the importance of the default mode network in subsequent memory for the temporal context of a clip, especially when the clip is part of a coherent narrative.

Reviewer #1

*On the previous round of review, my major concern was that the change attributed to putative contextual drift that correlates with duration judgments could more simply be attributed to perceptual/semantic differences in the patterns themselves. Real world stimuli that unfold in time are autocorrelated over just about every time scale. Compounding the problem, it is impossible to measure the similarity on all relevant dimensions. This revision makes a substantive attempt to argue against the perceptual hypothesis, there are two behavioral controls that attempt to address the question of whether the results attributed to contextual change could be driven by perceptual effects. To summarize my reaction to the revision, while the controls make for a stronger case than the previous submission, I am not convinced that these controls address the concern in a satisfactory way. I suggest additional analyses with the existing data that could clarify this point. If this concern were resolved (which is not at all clear) the manuscript would result in a very nice contribution.*

*The first control asks subjects to describe event boundaries during presentation of the story. This allows a rough estimate of the change in context between the two clips. Indeed the number of event boundaries predicted duration judgments by the original participants. The assumption seems to be that there is no perceptual/semantic similarity across event boundaries, but is not an assumption that I can accept. Imagine a story where Alice and Betty have a discussion at the beach. Then there's an event boundary and Alice and Betty move to the coffee shop. Then there's another event boundary as Alice leaves and Betty and Chris have a conversation in the coffee shop. The number of event boundaries correlates with the overlap of perceptual/semantic features present in the scene. More broadly, if the perceptual/semantic content is autocorrelated over long time scales (and it almost surely is) and if event boundaries are a proxy for abrupt drops in the autocorrelation, then number of event boundaries really ought to predict perceptual/semantic similarity of the available features. So this control is not at all convincing.*

We thank the reviewer for pointing out this important concern and completely agree that the number of event boundaries between two clips should correlate with the degree of perceptual and semantic dissimilarity between them. In fact, we discussed this possibility in the previous version of the manuscript but did not make it sufficiently explicit in our argument:

In the previous manuscript, we showed that the number of event boundaries between two clips was significantly more correlated with original duration estimates than with naïve duration estimates. Based on this result, we concluded that having memory of the story caused the original participants to be more influenced by the number of event boundaries. In other words, we hoped to infer that our original participants were influenced by their memory of events that had occurred between the two clips when estimating durations.

We have modified this section in order to better emphasize this logic and placed it in the Behavioral Results section of the revised manuscript:

“However, it is important to note that the number of event boundaries between two clips also influences the perceptual and semantic similarity between them (e.g., clips from the same scene might sound more similar than clips from different scenes). […] This suggests that the number of event boundaries carries information about temporal context that is not contained within the clips alone, and that our original participants’ estimates were influenced by their memory of this contextual information.”

In the second control, a group of naive subjects are asked to rate the similarity of the clips. There is no evidence that their ratings correspond to the number of event boundaries between the clips. The suggestion is that because number of event boundaries indexes contextual change (but not presumably perceptual/semantic similarity), the null result requires us to accept that there is no difference in the similarity of the two clips. Leaving aside for a moment the issue of asking the reader to accept the null (which is a really serious problem!), this is kind of an indirect test of what we're really after. The finding is that duration judgments of the fMRI subjects correlate with number of event boundaries whereas the judgments of the naive controls do not correlate with number of event boundaries. Why not just ask whether the judgments of the fMRI subjects correlate with the judgments of the naive subjects? If they do, then there is no way to argue that the change in the multivoxel signal is due to contextual drift per se. It might be possible to partial out the effect attributable to the naive subjects' judgments. If there is not a correlation, then the authors still have to successfully argue for the null, but it's at least a clean and direct (and much more sensitive!) test of the question of interest.

We thank the reviewer for highlighting the importance of discussing the correlation between original behavior and naïve behavior. In the previous manuscript, we did correlate the naïve duration judgments directly with the original duration judgments and reported the results:

“The inter-subject correlation in duration estimates was as strong for naïve participants (M=0.43, SD=0.18) as for our original participants (M=0.43, SD=0.25), suggesting that they used a consistent strategy to estimate durations. However, when we correlated duration estimates from our original group of participants with those of our naïve participants, we found that between-group correlations (M=0.18, SD=0.22) were significantly lower than the within-group correlations (p<0.0001, as assessed by a permutation test described in the Materials and methods). This suggests that while both groups used a consistent strategy to estimate durations, the nature of the strategy differed across groups.”

Using these results, we never meant to argue that *none* of the original duration estimates could be explained by the perceptual or semantic similarity of the clips. In other words, we never meant to argue for the null. In fact, we feel it would be surprising if the duration estimates did not correlate with perceptual and semantic similarity, since presumably a large component of mental context change is driven by changes in perceptual and semantic changes.

By showing that the within-group correlations were significantly stronger than the between-group correlations, we hoped only to show that the two groups were using qualitatively different strategies. Together with the significantly greater correlation between original estimates and event boundaries, we hope these results show that there is a component of the duration estimates that was driven by memory and that could not be explained by perceptual similarity alone.

To avoid a misinterpretation of the argument, we have made these claims more explicit in the revised manuscript. We have also added confidence intervals for the within-group and between-group correlations.

“When we correlated duration estimates from our original group of participants with those of our naïve participants, we found that the between-group correlations (M=0.18, SD=0.22, 95% CI=[0.04, 0.28]) were significantly above 0, suggesting that a component of the original duration estimates was influenced by the similarity in content between clips. However, the between-group correlations were significantly lower than the within-group correlations (p<0.0001, as assessed by a permutation test described in the Materials and methods). In other words, there is a reliable component of our original participants’ behavior that cannot be captured by accounting for the perceptual and semantic similarity between clips. In summary, having memory of the story induced a qualitatively different pattern of behavior and produced significantly more accurate duration estimates.”

Most importantly, we did not mean to argue that these behavioral results show anything about the neural data. In the previous manuscript, we attempted to delimit the implications of the behavioral data, and to point out that only the within-interval neural analysis enables us to rule out perceptual similarity as an explanation of the neural effects:

“These results suggest that duration estimates do not correlate with the number of contextual changes when participants are judging temporal distance based purely on the content of the clips. […]

However, it is still possible that pattern distance in the brain regions we found correlates with the component of duration estimates that is driven by the perceptual and semantic similarity between clips, rather than by contextual changes. To rule out this possibility, we performed a version of our main analysis that holds constant the perceptual and semantic similarity between two clips.”

In the within-interval analysis, we correlated individual differences in subjective duration for a given interval with individual differences in neural pattern distance for that interval. By performing the correlation within a given interval, we hold constant the perceptual and semantic content of the two clips and only leverage individual differences in how long the interval appeared retrospectively.

To better emphasize the importance of the within-interval analysis as a control for the objective similarity between two clips, we have placed the within-interval ROI analysis in the Results section of the revised manuscript, directly after the within-participant ROI analysis. We have also added a whole-brain searchlight version of the within-interval analysis, which we have placed directly after within-participant whole-brain searchlight, in the revised manuscript. Throughout the Results and Discussion of the revised manuscript, we have tried to highlight the importance of showing that a brain region is significant in both the within-participant analysis, which controls for subject random effects, and the within-interval analysis, which controls for item random effects. This is particularly evident in the section entitled “Comparing Results from ROI and Searchlight Analyses” (pp. 33-35), where we highlight that the right entorhinal cortex was the only ROI that survived both types of analyses, whereas the searchlight clusters from both types of analyses overlapped in areas like the right amygdala, right temporal pole and right posterior parahippocampal gyrus.

Mixed-Effects Modeling

In addition to the within-interval analysis, we sought to more thoroughly address the concern that patterns of activity in the regions we found might represent the perceptual or semantic content of the clips, rather than abstract contextual information. For this purpose, we fit a mixed-effects model to the data from each ROI, controlling for the effect of naïve duration estimates. For each ROI, we fit a model of this form:

*SubjectiveDuration ~ 1 + NeuralDistance + NaiveDuration + (1 + NeuralDistance | Interval) + (1 + NeuralDistance + NaiveDuration | Subject)*

As described in the revised manuscript,

“This analysis estimates population-level effects of interest, while controlling for the possibility of individual variability between subjects and between clip pairs. In other words, this approach leverages the power of the within-interval analysis to control for the objective content similarity between two clips, while also taking into account variability in the effect across participants. In addition, we included the mean duration estimates from our naïve participants as a covariate in the model (see Behavioral Results). Since naïve participants had estimated the temporal distance between each pair of clips without hearing the story, this covariate is a further control for the inherent guessability of the temporal distance between two clips. Both controls strengthen our interpretation that the remaining effect of neural pattern distance on duration estimates is driven by the contextual dissimilarity (rather than perceptual or content dissimilarity) between two clips.”

Out of the 84 anatomical ROIs, we found that the fixed effect of neural pattern distance on duration estimates was positive (i.e., had bootstrapped confidence intervals that did not include 0) in the right entorhinal cortex and left caudal anterior cingulate cortex (ACC). We also found near-significant effects in the right amygdala and right superior temporal cortex. In this model, a significant fixed effect means that the effect generalizes to the population, even after variability across participants and intervals, as well as the other covariates (i.e., naïve duration estimates) have been accounted for.

Importantly, including the naïve duration estimates in the model did not have a significant impact on the size of the fixed effect in these regions, suggesting that the relationship between neural distance and duration estimates was not driven solely by the perceptual or semantic dissimilarity between clips. These results are detailed in subsection “Mixed-Effects Model Accounting for Naïve Duration Estimates” of the revised manuscript.

Reviewer #2

*This revision has been responsive to prior concerns about whether visual or semantic dissimilarity during temporal memory judgments could be used to infer how far apart the clips had been presented during encoding. The authors conducted behavioral analyses to show that distance judgments were related to listening to the story and could not be deduced based on the test stimuli alone. They show that the number of event boundaries experienced in between the test stimuli also modulated distance judgments. These new results remove any doubts that the reported effects are not driven by visual confounds.*

*However, I am somehow not that excited about the new ms as it now provides a list of more regions that show pattern change related to distance judgments – much of the medial temporal lobe, frontal cortex, anterior temporal cortex, ACC… given the effects are more widespread, the laser focus on entorhinal and pars orbitalis makes the paper not easy to digest. Is this a general broad signal? Or is it focused?*

We thank the reviewer for pointing out this inconsistency between the breadth of our results and the focus on two specific regions in our discussion. Our Results section has undergone substantial revisions, which now make it easier to compare and synthesize the results across analyses.

We have restructured the Results section to have the following sub-sections:

1) Within-participant ROI analysis

2) Within-interval ROI analysis

3) Mixed-Effects ROI analysis

4) Within-participant Searchlight analysis

5) Within-interval Searchlight analysis

6) Comparing ROI and Searchlight analyses

Sub-section 1 reports significant effects in the right entorhinal, right pars orbitalis and left caudal ACC. Sub-section 2 reports significant effects in the right entorhinal, right amygdala and right insula. Sub-section 3 reports significant effects in the right entorhinal cortex and left caudal ACC. Sub-sections 4 and 5 both report significant clusters in the right anterior temporal lobe.

Sub-section 6 summarizes all the above results and highlights the fact that only the right entorhinal cortex reliably survived all versions of the ROI analysis, whereas the searchlight clusters overlap with this region as well as parts of the right amygdala, temporal pole, anterior middle temporal gyrus and posterior parahippocampal gyrus. Thus, our final results are localized to the right anterior temporal lobe, peaking in the right entorhinal cortex.

Please note that the results for the within-interval ROI analysis (sub-section 2) have changed very slightly in the revised manuscript. When we were reproducing these results, we noticed a mistake in the MATLAB code, which used a slightly different threshold to determine which intervals were labeled as “confident”. In the rest of the paper, we ensured that the confidence threshold for each participant would keep at least *1/3* of the behavioral data. However, for the within-interval analysis, we had mistakenly used a confidence threshold for each participant that would keep at least *1/2* of the behavioral data. Thus, the within-interval analysis was mistakenly using more of the behavioral data (a less stringent confidence threshold) than the other analyses in the paper. Correcting this error resulted in a very minor change in the Z-values for this analysis, though this change was sufficient to bring several of the ROIs below the q<0.05 FDR threshold. Using the correct confidence threshold, the new results show that only the right entorhinal, right amygdala and right insula pass FDR correction (q<0.05) for the within-interval ROI analysis, and the right entorhinal cortex even survives whole-brain correction (among 84 anatomical regions) at q<0.05.

Also new data that has been added in response to other concerns that now raise some skepticism. The most intrusive is the fact that distance judgments vary predictably by 'list' position – events early in the audiovisual recording are remembered as farther apart than those later in the tape. (see Figure 11 – top panel). However, pattern similarity estimates do not track this behavioral effect. This result raises questions about why, if entorhinal and frontal cortex are representing temporal context signal, would they not also some what mirror the behavioral judgments. I could imagine that context representations may play less of a role as item memory fades? This is not discussed but should be for the authors' views on this to be clear. Otherwise, the impact of the final result is unclear.

We thank the reviewer for pointing out the importance of investigating this discrepancy between the neural and behavioral data. We have modified this section of the manuscript and included an extensive discussion of the possible reasons for this discrepancy.

First, we discuss the positive time-order effect, the canonical finding that duration estimates are larger at the start of a new “episode” and decrease over time within the episode (i.e., duration estimates might be longer at the beginning of the story because context changes more at the start of a novel episode). Second, we show that there are significantly more event boundaries in the beginning of our story, and that the number of event boundaries decreases with time in story. Both of these factors might explain why duration estimates decrease with time in story.

Importantly, we then discuss why neural pattern change did not decrease over time in the story. Since this trend was not present in any of the regions uncovered by our ROI analyses (right entorhinal, right OFC, left caudal ACC, right amygdala, right insula), we performed a whole-brain search, to check whether any anatomical region exhibits this decrease in pattern change over time. Surprisingly, we did not find any region exhibiting this pattern significantly. Given that we were looking for a slow change in neural signal (unfolding over the entire time course of the story), we thought that our high-pass filter might be removing this slow change; to address this possibility, we analyzed the unfiltered data. When we did this, we found an overwhelming trend in the opposite direction, with most brain patterns changing *more* with time in the experiment. This increase in pattern change over time was even present in the CSF and white matter, suggesting that it was not reflective of neuronal activity, but was probably caused by a non-neuronal artifact, such as scanner drift or motion, that increased slowly with time.

In conclusion, we argue that even if neural activity patterns were changing less and less as the story unfolds, in concert with the behavior, we might not be able to see this effect, as it would have to overcome a global signal in the opposite direction that is not due to neural activity and is present everywhere, including the CSF.

*In my original review, I had requested that they examine whether univariate activity was related to temporal memory success. The did run that analysis but only in entorhinal cortex and pars orbitalis -and do not see any effects but beg off reporting it since they did not have a priori predictions about it. however, published work (Jenkins and Ranganath) has shown that univariate activity in more dorsal parts of lateral frontal cortex is related to coarse temporal memory judgments. so there is a clear precedent for this effect. I am not sure why they say they did not have that prediction but this analysis, even if the results DO NOT show a univariate effect would be informative and could even bolster their conclusions that patterns across time, rather than activity to any single event, is a better predictor of temporal memory judgments.*

We apologize for not having addressed the reviewer’s question more thoroughly in our previous round of revisions, and we appreciate the suggestion of performing a whole-brain analysis to search for the relationship found by Jenkins and Ranganath (2010) between univariate activity at encoding and subsequent coarse temporal memory judgments.

In addressing this request, we found that (in our previous analysis) we had quantified the accuracy of participants’ timeline judgments in a slightly different manner from the Jenkins and Ranganath (2010) analysis. Jenkins and Ranganath (2010) had performed a linear regression of estimated temporal position against actual temporal position, and used the absolute value of the residuals as their measure of “error”. In contrast, we had used the absolute value of the distance between the estimated place in the story and the actual place in the story as our measure of error. Using this absolute distance method, we reported null effects in the right entorhinal cortex and right pars orbitalis, and in fact we did not find significant effects in any ROI in the brain.

In a new version of this analysis, we have now followed the Jenkins and Ranganath procedure more closely by performing a linear regression on the behavior. We then correlated the negative of the error (our measure of accuracy) with the activity of each brain voxel at encoding. We found highly significant clusters in the left dorsolateral prefrontal cortex (replicating the above report), as well as the medial prefrontal cortex, and slightly sub-threshold clusters in the medial parietal (precuneus and retrosplenial) and left superior temporal gyrus (Figure 14, left panel, blue clusters). Thus, it seems that the linear regression procedure mattered for the final results.

Importantly, in this analysis, we performed a full correlation between a voxel’s activity when a clip was encoded and the participant’s accuracy in placing that clip on the timeline. Jenkins and Ranganath had binarized the behavior into “Hits” (bottom 1/3 of residual errors) and “Misses” (top 1/3 of residual errors). When we binned the behavior in a similar way, we found that the medial parietal, medial prefrontal and left superior temporal clusters were all highly significant, whereas the left dorsolateral PFC cluster was no longer significant (Figure 14, right panel, red-yellow clusters).

Author response image 1.Clusters whose activity at encoding correlated with subsequent accuracy at placing clips on the timeline of the story.The left panel (in blue) shows the results of the Pearson’s correlation between accuracy on a clip and the encoding activity for that clip. The right panel (red-yellow) shows the results of the contrast between activity for Hits (bottom 1/3 of residual error) and activity for Misses (top 1/3 of residual error) when placing the clip on the timeline of the story.**DOI:**
http://dx.doi.org/10.7554/eLife.16070.028

In subsection “Replication of Jenkins and Ranganath 2010: activity at encoding predicts accuracy of temporal context memory”, we report the analysis where a full correlation between voxel activity and behavioral accuracy is performed (rather than the contrast between Hits and Misses). We do not report both versions for the sake of brevity.

To summarize, both versions of the whole-brain analysis revealed that regions of the Default Mode Network (DMN) were significantly more active during the encoding of clips whose place in the timeline of the story participants later recalled more accurately.

In the manuscript’s discussion, we propose that one reason for the discrepancy between the Jenkins & Ranganath results and our results may be due to the narrative structure of our stimulus, which seems to elicit strong inter-subject correlations in regions of the DMN (Lerner et al., 2011). It is possible that this network is particularly important for encoding the temporal context of stimuli that are part of a narrative (Chen et al., 2015), but that another strategy is used when the stimuli whose timing is recalled were not related to one another.

Regarding the reviewer’s comment:

*“even if the results DO NOT show a univariate effect would be informative and could even bolster their conclusions that patterns across time, rather than activity to any single event, is a better predictor of temporal memory judgments.”*

The reviewer is suggesting that the results of the univariate analysis could potentially bolster our conclusions from the multivariate analysis. However, please note that it is not necessarily surprising that our multivariate analyses (which constitute the bulk of the manuscript) reveal different results from this univariate analysis, given that the behavioral tests used are different. In the bulk of the manuscript, we relate multivariate pattern change to duration estimates, where participants explicitly estimated the relative distance between two clips. On the other hand, the univariate analysis leverages data from a separate behavioral test where participants placed each clip, individually, on the timeline of the story (not a comparison between two clips).